# What I Cannot Predict, I Do Not Understand:
# A Human-Centered Evaluation Framework for
# Explainability Methods

**Julien Colin**[1,3,5 *†]     **Thomas Fel**[1,3,4 *]     **Rémi Cadène**[1,2 ‡]     **Thomas Serre**[1,3]

[1]Carney Institute for Brain Science, Brown University, USA   [2]Sorbonne Université, CNRS, France
[3]Artificial and Natural Intelligence Toulouse Institute, Université de Toulouse, France
[4] Innovation & Research Division, SNCF
[5] ELLIS Alicante, Spain
{julien_colin,thomas_fel,remi_cadene}@brown.edu

## Abstract

A multitude of explainability methods has been described to try to help users better understand how modern AI systems make decisions. However, most performance metrics developed to evaluate these methods have remained largely theoretical – without much consideration for the human end-user. In particular, it is not yet clear (1) how useful current explainability methods are in real-world scenarios; and (2) whether current performance metrics accurately reflect the usefulness of explanation methods for the end user. To fill this gap, we conducted psychophysics experiments at scale ($n = 1,150$) to evaluate the usefulness of representative attribution methods in three real-world scenarios. Our results demonstrate that the degree to which individual attribution methods help human participants better understand an AI system varies widely across these scenarios. This suggests the need to move beyond quantitative improvements of current attribution methods, towards the development of complementary approaches that provide qualitatively different sources of information to human end-users.

## 1   Introduction

There is now broad consensus that modern AI systems might not be safe to be deployed in the real world [1] despite their exhibiting very high levels of accuracy on held-out data because these systems have been shown to exploit dataset biases and other statistical shortcuts [2–5]. A growing body of research thus focuses on the development of explainability methods to help better interpret these systems' predictions [6–14] to make them more trustworthy. The application of these explainability methods will find broad societal uses, like easing the debugging of self-driving vehicles [15] and helping to fulfill the "right to explanation" that European laws guarantee to its citizens [16].

The most commonly used methods in eXplainable AI (XAI) are attribution methods, but despite a plethora of different approaches [6–10, 12, 17–19], assessing the quality and reliability of these methods remains an open problem. So far the community has mostly focused on evaluating these methods using surrogate measures defined axiomatically [15, 20–23]. The two most popular approaches are: (1) using ground truth annotations [9, 10, 12, 24–27] and (2) measuring *faithfulness* using objective metrics  [9, 12, 24, 25, 28, 29].

The first approach takes humans into consideration only in so far as it evaluates the alignment of an explanation with a human annotation. This metric assumes that the classifier relies on human-

---

* Equal contribution  † Work done while the author was working at Brown University  ‡ Work done before April 2021 and joining Tesla

like features[*], which is an assumption that turns out to be erroneous [2–4], and unsurprisingly the results obtained on those benchmarks do not correlate with the ability of attribution methods to help humans in decision making tasks [30]. On the other hand, the second approach focuses purely on the relationship between the explanations and the model, i.e., they assess whether the explanations actually reflect the evidence for the prediction. They have emerged as the primary means of evaluating attribution methods, but because they take the human out-of-the-loop of the evaluation, it is not clear how well they can predict the practical usefulness of attribution methods and their possible failures.

As previously highlighted in [30], relying solely on current theoretical benchmarks to evaluate attribution methods can be a risky endeavor as they are totally detached from humans. Indeed, the seminal work of [31] has emphasized keeping in mind the end goal of XAI: to develop useful methods, i.e., methods that help the user better understand their model. Thus, they recommend systematically having recourse to human experiences. Moreover, a large body of research from the psychology literature has endorsed the argument that the functional role of explanations is to support learning and generalization [32–38] – i.e., can an explanation help identify generalizable rules that readily transfer to unseen instances? Therefore, evaluating attribution methods on their ability to help users understand general rules to predict a classifier decisions is perfectly aligned with this vision.

To that end, [31] suggest evaluating explainability methods directly through the end goals of XAI. We subscribe to this idea and, in this work, consider three main real-world scenarios that illustrate potential applications of explainability methods: **(1)** identifying a potential bias in a system's decision along with its source using the classic Husky vs. Wolf dataset [6] (previous work has documented the risk associated with a biased model [39–41]); **(2)** identifying novel strategies discovered by a system [42, 43] for tasks that would be nearly impossible for humans [44–47], here we consider a highly complex categorization problem for non-expert humans [48, 49]; and **(3)** understanding failure cases [4, 39] on a subset of ImageNet [50].

The main contributions of this paper are as follows:

- We propose a novel human-centered explainability performance measure together with associated psychophysics methods to experimentally evaluate the practical usefulness of modern explainability methods in real-world scenarios.

- Large-scale psychophysics experiments ($n = 1,150$) revealed that SmoothGrad [8] is the most useful attribution method amongst all tested and that none of the faithfulness performance metrics appear to predict if and when attribution methods will be of practical use to a human end-user.

- Perceptual scores derived from attribution maps, characterizing either the complexity of an explanation or the challenge associated with identifying "what" features drive the system's decision, appear to predict failure cases of explainability methods better than faithfulness metrics.

## 2 Related work

**Evaluations based on faithfulness measures** Common approaches [9, 28] measure the faithfulness of an explanation through the change in the classification score when the most important pixels are progressively removed. The bigger the drop, the more faithful is the explanation method. To ensure that the drop in score does not come from a change in the distribution of the perturbed images, the ROAR [51] methods include an additional step whereby the image classifier is re-trained between each removal step. Because these methods do not require ground-truth annotations (i.e. object masks or bounding boxes), they are quite popular in computer vision [9, 12, 24, 25, 28, 29] and natural language processing [12, 52, 53].

Nevertheless, *faithfulness* measures have recently been criticized as they all rely on a baseline for removing important areas, a baseline that will obviously give better scores to methods relying internally on the same baseline [54]. More importantly, they do not consider humans at any time in the evaluation. As a result, it is unclear if the most faithful attribution method is practically useful to humans.

**Evaluations based on humans** A second class of approaches consists in evaluating the ability of humans to leverage explanations for different purposes [6, 10, 30, 55–62]. [6] were the first to evaluate the usefulness of explanations. Their work focused on the use case of bias detection: they

---

[*]One advantage of our approach is that it is independent of the classifier, whether it relies on human-like features or it is even correct.

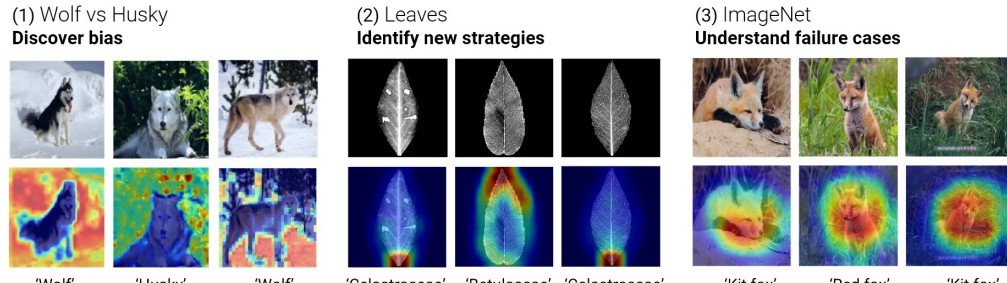

| (1) Wolf vs Husky | (2) Leaves | (3) ImageNet |
| **Discover bias** | **Identify new strategies** | **Understand failure cases** |

'Wolf'  'Husky'  'Wolf'  'Celastraceae'  'Betulaceae'  'Celastraceae'  'Kit fox'  'Red fox'  'Kit fox'

Figure 1: We study the practical usefulness of recent explainability methods in three real-world scenarios, each corresponding to different use cases for XAI. The first dataset is Husky vs. Wolf where the goal of the explanations is to help the user to identify a source of bias in a model (classification is based on the background (snow, grass) as opposed to the animal). The second dataset corresponds to a real-world leaf classification problem which is complex for non-experts. The goal of the explanations is to help the end-user identify the strategy discovered by the vision system. Finally, the third dataset is a subset of ImageNet, which consists of a collection of images where half have been misclassified by the system. The goal of the explanations here is to help the end-user understand the failure sources of a high performing model.

trained a classifier on a biased dataset of wolves and huskies and found that the model consistently used the background to classify. They asked participants if they trusted the model before and after seeing the explanation for the model's predictions, and found that explanations helped detect bias here. We use a similar dataset to reproduce those results, but our evaluation differs greatly from theirs as we do not ask if participants trust the model but instead measure directly if they understand it.

Closest to our work are [59–61, 63, 64]. [59–61] design their evaluation around the notion of *simulatability* [31, 65]. They introduce different experimental procedures to measure if humans can learn from the explanations how to copy the model prediction on unseen data. Some provide the explanations at test time [59, 60]. Similar to us but for tabular data, [61] proposes to hide explanations at test time, this forces the participants to learn the rules driving the model's decision at training time where the explanations are shown. There are two limitations to their work: (1) they provide ground-truth labels associated with input images during training, (2) the participants see the same set of images without explanations, and then with explanations, always in that order. This creates learning effects that can heavily bias their results. We differ from their work by: (1) removing ground-truth labels from our framework as they serve no purpose and can bias participants, and (2) we have different participants go through the different conditions. This removes any learning effect, and more importantly, new explainability methods can be evaluated independently and still be compared to the previously evaluated methods. A recent study [63] evaluated how AI systems may be able to assist human decisions by asking participants to identify the correct prediction out of four prediction-explanations pairs shown simultaneously. This measure reflects how well explanations help users tell apart correct from incorrect predictions. While the approach was useful to evaluate explanations in this specific scenario, it is not clear how this framework could be used to evaluate explainability methods more generally. Furthermore, when comparing different types of methods, they adapt the complexity of certain explainability methods to ease the task for participants. We argue that the complexity of explanations is an important property of explanations and that abstracting it away from the evaluation lead to unfair comparisons between methods. In contrast, we propose a more general evaluation framework that can be used for any kind of explainability method without the need to adapt them for the evaluation procedure – hence allowing for an unbiased and scalable comparison between methods. Finally, [64] proposes to evaluate if users are able to identify important features biasing the predictions of a model using a synthetic dataset. By controlling the generation process of the dataset, they have access to the ground-truth attributes biasing the classifier, and can measure the accuracy of users at identifying these features. They evaluate if concept-based or counterfactual explanations help users improve over a baseline accuracy when no explanations are provided, and find no explanation tested to be useful. While both works highlights the importance of human evaluation, they differ in: the metrics employed (identifying relevant features for the model *vs.* meta-prediction), the type of dataset used (synthetic *vs.* real-world scenarios), and the type of methods evaluated (counterfactual and concept-based methods *vs.* attribution methods).

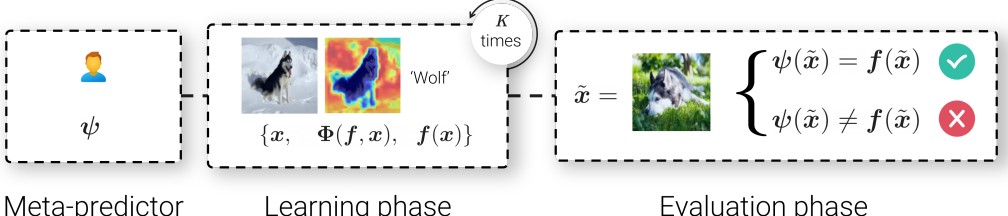

Meta-predictor     Learning phase          Evaluation phase

Figure 2: We describe a human-centered framework to evaluate explainability methods borrowing the concept of Meta-predictor. The framework requires a black box model $\boldsymbol{f}$ (the predictor), an explanation method $\boldsymbol{\Phi}$ and a human subject $\psi$ which will try to predict the predictor, hence, the name Meta-predictor. The first step is the learning phase where the Meta-predictor is training using $K$ samples $\boldsymbol{x}$, together with the associated model predictions $\boldsymbol{f}(\boldsymbol{x})$ and explanations $\boldsymbol{\Phi}(\boldsymbol{f}, \boldsymbol{x})$. The goal of this learning phase is for the Meta-predictor to uncover the rules driving the decisions of the model from the triplets $(\boldsymbol{x}, \boldsymbol{\Phi}(\boldsymbol{f}, \boldsymbol{x}), \boldsymbol{f}(\boldsymbol{x}))$. Then, the second step is the evaluation phase where we test the Meta-predictor's ability to correctly predict the model's outputs on new samples $\tilde{\boldsymbol{x}}$ by comparing its predictions $\psi(\tilde{\boldsymbol{x}})$ to those of $\boldsymbol{f}(\tilde{\boldsymbol{x}})$. The *Utility* score of the explanation method is then computed as the relative accuracy improvement of Meta-predictor trained with vs. without explanations.

## 3 Proposed evaluation framework

Before providing a rigorous definition of interpretability, let us motivate our approach with an example: a linear classifier is often considered to be readily interpretable because its inner working is sufficiently intuitive that it can be comprehended by a human user. A user can in turn build a mental model of the classifier – predicting the classifier's output for arbitrary inputs. In essence, we suggest that the model is interpretable because the output can be predicted – i.e, we say we understand the rules used by a model, if we can use those inferred rules to correctly predict its output. This concept of predicting the classifier's output is central to our approach and we conceptualize the human user as a *Meta-predictor* of the machine learning model. This notion of Meta-predictor is also closely related to the notion of *simulatability* [24, 31, 61, 65, 66]. We will now define the term more formally.

We consider a standard supervised learning setting where $\boldsymbol{f}$ is a black-box predictor that maps an input $\boldsymbol{x} \in \mathcal{X}$ (e.g., an image) to an output $\boldsymbol{f}(\boldsymbol{x}) \in \mathcal{Y}$ (e.g., a class label). One of the main goals of eXplainable AI is to yield useful rules to understand the inner-working of a model $\boldsymbol{f}$ such that it is possible to infer its behavior on unseen data points. To correctly infer those rules, the usual approach consists in studying explanations (from Attribution Map, Concept Activation Vectors, Feature Visualization, *etc*..) for several predictions. Formally, $\boldsymbol{\Phi}$ is any explanation functional which, given a predictor $\boldsymbol{f}$ and a point $\boldsymbol{x}$, provides an information $\boldsymbol{\Phi}(\boldsymbol{f}, \boldsymbol{x})$ about the prediction of the predictor. In our experiments, $\boldsymbol{\Phi}$ is an attribution method but we would like to remind that the framework is naturally adaptable to other explainability methods such as concept-based methods or feature visualization.

**The understandability-completeness trade-off** Different attribution methods will typically produce different heatmaps – potentially highlighting different image regions and/or presenting the same information in a different format. The quality of an explanation can thus be affected by two factors: *faithfulness* of the explanation (i.e., how many pixels or input dimensions deemed important effectively drive the classifier's prediction) and the understandability of the explanation for an end-user (i.e., how much of the pattern highlighted by the explanation is grasped by the user).

At one extreme, an explanation can be entirely *faithful* and provide all the information necessary to predict how a classifier will assign a class label to an arbitrary image (i.e., by giving all the parameters of the classifiers). However, such information will obviously be too complex to be understood by a user and hence it is not *understandable*. Conversely, an explanation that overly simplifies the model might offer an approximation of the rule used by the model that will be more easily grasped by the user –a more *understandable* explanation– but this approximation might ultimately mislead the user if it is not *faithful*. That is to say, just because a human agrees with the evidence pointed out by an explanation does not necessarily mean that it reflects how the model works.

Overall, this means that there is a trade-off between the amount of information provided by an explanation and its comprehensibility to humans. The most useful explanations should lie somewhere in the middle of this trade-off.

**The *usefulness* metric** We describe a new human-centered measure that incorporates this trade-off into a single *usefulness* measure by empirically evaluating the ability of human participants to learn to "predict the predictor", i.e., to be an accurate Meta-predictor. Indeed, if an explanation allows users to infer precise rules for the functioning of the predictor on past data, the correct application of these same rules should allow the user to correctly anticipate the model's decisions on future data. Scrutable but inaccurate explanations will result in an inaccurate Meta-predictor – just like accurate inscrutable ones. This Meta-predictor framework avoids current pitfalls such as confirmation bias - just because a user likes the explanation does not mean they will be a better Meta-predictor - or prediction leakage on the explanation - in simulatability experiments, as the explanation is available during the test phase, any explanation that leaks the prediction would have a perfect score, without giving us any additional information about the model. We will now formally describe the metric build using this framework.

We assume a dataset[†] $\mathcal{D} = \{(\boldsymbol{x}_i, \boldsymbol{f}(\boldsymbol{x}_i), \boldsymbol{\Phi}(\boldsymbol{f}, \boldsymbol{x}_i)\}_{i=1}^K$ used to train human participants to learn to predict a classifier's output $\boldsymbol{f}$ from $K$ samples made of an input image $\boldsymbol{x}_i$, the associated predictions $\boldsymbol{f}(\boldsymbol{x}_i)$ and explanations $\boldsymbol{\Phi}(\boldsymbol{f}, \boldsymbol{x}_i)$. We denote $\boldsymbol{\psi}^{(K)}$ a human Meta-predictor after being trained on the dataset $\mathcal{D}$ (see Fig. 2) using explanations. In addition, let $\boldsymbol{\psi}^{(0)}$ be the human Meta-predictor after participants were trained on the same dataset but without explanations to offer baseline accuracy scores. We can now define the usefulness of an explainability method $\boldsymbol{\Phi}$ after training participants on $K$ samples through the accuracy score of the Meta-predictor normalized by the baseline Meta-predictor accuracy:

$$Utility\text{-}K = \frac{\mathbb{P}(\boldsymbol{\psi}^{(K)}(\boldsymbol{x}) = \boldsymbol{f}(\boldsymbol{x}))}{\mathbb{P}(\boldsymbol{\psi}^{(0)}(\boldsymbol{x}) = \boldsymbol{f}(\boldsymbol{x}))} \tag{1}$$

with $\mathbb{P}(\cdot)$ the probability over a test set. Thus, *Utility-K* score measures the improvement in accuracy that the explanation has brought. It is important to emphasize that this *Utility* measure only depends on the classifier prediction and not on the ground-truth label as recommended by [67]. After fixing the number of training samples $K$, we compare the normalized accuracy of different Meta-predictors. The Meta-predictor with the highest score is then the one whose explanations were the most useful as measures compared to a no-explanation baseline.

***Utility* metric** In practice, we propose to vary the number of observations $K \in \{K_0, ..., K_n\}$ and to report an aggregated *Utility* score by computing the area under the curve (AUC) of the *Utility-K*. The higher the AUC the better the corresponding explanation method is. Formally, given a curve represented by a set of $n$ points $\mathcal{C} = \{(K_0, Utility\text{-}K_0), ..., (K_n, Utility\text{-}K_n)\}$ where $K_{i-1} < K_i$ we define the metric as $Utility = AUC(\mathcal{C})$.

## 4 Experimental design

We first describe how participants were enrolled in the study, then our general experimental design (See SI for more informations).

**Participants** Behavioral data were gathered from $n = 1,150$ participants using Amazon Mechanical Turk (AMT) (www.mturk.com). All participants provided informed consent electronically and were compensated \$1.4 for their time ($\sim 5-8$ min). The protocol was approved by the University IRB and was carried out in accordance with the provisions of the World Medical Association Declaration of Helsinki. For each of the three tested datasets, we ensured that there was a sufficient number of participants after filtering out uncooperative participants ($n = 240$ participants, 30 per condition, 8 conditions) to guarantee sufficient statistical power (See SI for details). Overall, the cost of evaluating one method using our benchmark is relatively modest (\$50 per test scenario).

**General study design** It included 3 conditions: an experimental condition where an explanation is provided to human participants during their training phase (see Fig. 2), a baseline condition where no explanation was provided to the human participants, and a control condition where a bottom-up saliency map [68] was provided as a non-informative explanation. This last control is critical, and indeed lacking from previous work [6, 61], because it provides a control for the possibility that

---

[†]We note that, in this paper, we only considered binary dataset –Class 1 vs Class 2– because having the participants classify more than 2 classes would increase their cognitive load and bring unnecessary difficulty to the task. Nonetheless, any dataset could have been used as classification problems with more than 2 classes can always be trivially reformulated as Target class vs. Other / binary classification problems, instead of Class 1 vs Class 2, without lack of generality.

| Method | Husky vs. Wolf | | | | Leaves | | | | ImageNet | | | |
|---|---|---|---|---|---|---|---|---|---|---|---|---|
| Session n° | 1 | 2 | 3 | Utility | 1 | 2 | 3 | Utility | 1 | 2 | 3 | Utility |
| Baseline | 55.7 | 66.2 | 62.9 | | 70.1 | 76.8 | 78.6 | | 58.8 | 62.2 | 58.8 | |
| Control | 53.3 | 61.0 | 61.4 | 0.95 | 72.0 | 78.0 | 80.2 | 1.02 | 60.7 | 59.2 | 48.5 | 0.94 |
| Saliency [17] | 53.9 | 69.6 | 73.3 | 1.06 | 83.2 | 88.7 | 82.4 | **1.13** | 61.7 | 60.2 | 58.2 | 1.00 |
| Integ.-Grad. [7] | 67.4 | 72.8 | 73.2 | 1.15 | 82.5 | 82.5 | 85.3 | **1.11** | 59.4 | 58.3 | 58.3 | 0.98 |
| SmoothGrad [8] | 68.7 | 75.3 | 78.0 | **1.20** | 83.0 | 85.7 | 86.3 | **1.13** | 50.3 | 55.0 | 61.4 | 0.93 |
| GradCAM [10] | 77.6 | 85.7 | 84.1 | **1.34** | 81.9 | 83.5 | 82.4 | 1.10 | 54.4 | 52.5 | 54.1 | 0.90 |
| Occlusion [18] | 71.0 | 75.7 | 78.1 | **1.22** | 78.8 | 86.1 | 82.9 | 1.10 | 51.0 | 60.2 | 55.1 | 0.92 |
| Grad.-Input [69] | 65.8 | 63.3 | 67.9 | 1.06 | 76.5 | 82.9 | 79.5 | 1.05 | 50.0 | 57.6 | 62.6 | 0.95 |

Table 1: ***Utility-$K$* and *Utility* scores**. *Utility-$K$* scores across the 3 sessions for each attribution method, for each of the 3 datasets considered, followed by the *Utility* scores. Higher is better. The *Utility* scores of attribution methods that are statistically significant are **bolded**.

providing explanations along with training images simply increases participants' engagement in the task. As we will show in Sec. 5, such non-informative explanations actually led to a decrease in participants' ability to predict the classifier's decisions – suggesting that giving a wrong explanation is worse than giving no explanations at all.

Each participant was only tested on a single condition to avoid possible experimental confounds. The main experiment was divided into 3 training sessions (with 5 training samples in each) each followed by a brief test. In each individual trial, an image was presented with the associated prediction of the model, either alone for the baseline condition or together with an explanation for the experimental and control condition. After a brief training phase (5 samples), participants' ability to predict the classifier's output was evaluated on 7 new samples during a test phase. During the test phase, no explanation was provided to limit confounding effects: one possible effect is if the explanation leaks information about the class label.[‡] We also propose to use a reservoir that subjects can refer to during the testing phase to minimize memory load as a confounding factor which was reported in [61] (see SI for an illustration).

**Datasets and models**    We performed three distinct experiments in total – using a variety of neural network architectures and 6 representative attributions methods. Each of these experiments aimed at testing the usefulness of the explanation in a different context.

Our first scenario focuses on the detection of biases in AI systems using the popular Wolf vs. Husky dataset from [6] where an evaluation measure was already proposed around the usefulness of explanations for humans to detect biases. This makes it a good control experiment to measure the effectiveness of the framework proposed in Sec. 3. For this first experiment, we used the same model as in the original paper: InceptionV1 [70], and a similar dataset of Husky and Wolf images to bias the model. In this situation where prior knowledge of subjects can affect their Meta-predictor score, we balance data correctness ($50\%$ of correct/incorrect examples shown). Therefore, a subject relying only on their prior knowledge will end up as a bad Meta-predictor of the model. For this experiment, the results come from $n = 242$ subjects who all passed our screening process.

In our second scenario, we focus on a representative challenging image categorization task which would be hard to solve by a non-expert untrained human participant and the goal is for the end-user to understand the strategy that was discovered by the AI system. Here, we chose the leaf dataset described in [48]. We selected 2 classes from this dataset (Betulaceae and Celastracea) that could not be classified by shape to reduce the chances that participants will discover the solution on their own – forcing them instead to rely on non-trivial features highlighted by the explanations (veins, leaf margin, etc). This scenario is far from being artificial as it reflects a genuine problem for the paleobotanist [49]. Can explainability methods help non-specialists discover the strategies discovered by an AI system? As participants are lay people from Amazon Mechanical Turk we do not expect them to be experts in botany, therefore we did not explicitly try to control for prior knowledge. In this experiment, $n = 240$ subjects passed all our screening and other filtering processes.

---

[‡]Imagine an attribution method that would solely encode the classifiers' prediction. Participants would be able to guess the classifier's prediction perfectly from the explanation but the explanation per se would not help participants understand how the classifier works.

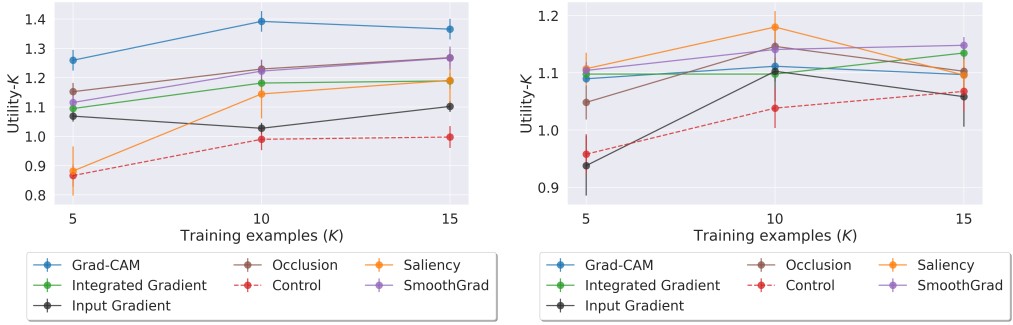

Figure 3: *Utility-$K$ for both Husky vs. Wolf (left) and the Leaves (right) dataset.* The *Utility-$K$* of the explanation, or the accuracy of the human Meta-predictor after training, is measured after each training session (3 in total) for the scenario (1) of bias detection (on the left) and the scenario (2) concerning the identification of new strategies. Concerning the first scenario, all methods have a positive effect on the score obtained - they improve the subjects' ability to predict the model - and are thus useful to better understand the model. Grad-CAM, Occlusion and SmoothGrad are particularly useful for bias detection. On the Leaves dataset [48], explanations are also useful, but specifically Saliency, SmoothGrad and Integrated Gradients.

Finally, our last scenario focuses on identifying cases where an AI system fails[§] using ImageNet [50], also used in previous explainability work [12, 24, 30, 51, 59, 71]. We used this dataset because we expect it to be representative of real-world scenarios where it is difficult to understand what the model relies on for classification which makes it very difficult to understand these failure cases. Moreover, previous work has pointed out that attribution methods are not useful on this dataset [59], we have thus chosen to extend our analysis to this particular case. We use a ResNet50 [72] pretrained on this dataset as predictor. Because prior knowledge is a major confounding factor on ImageNet, we select a pair of classes that was heavily miss-classified by the model, to be able to show subjects 50% of correct/incorrect predictions: the pair Kit Fox and Red Fox fits this requirement. In this experiment, we analyzed data from $n = 241$ participants who passed our screening and filtering processes.

For all experiments, we compared 6 representative attribution methods: Saliency (SA) [17], Gradient ⊙ Input (GI) [19], Integrated Gradients (IG) [7], Occlusion (OC) [18], SmoothGrad (SG) [8] and Grad-CAM (GC) [10]. Further information on these methods can be found in SI. Table 1 summarizes all the results from our psychophysics experiments.

## 5   Results

**Scenario (1): Bias detection**    Fig. 3 shows the *Utility-$K$* scores for each method after different numbers of training samples were used to train participants for the biased dataset of Husky vs. Wolf. The *Utility* score encodes the quality of the explanations provided by a method, the higher the score, the better the method, with the baseline score being 1 (every score is divided by the baseline score corresponding to human accuracy after training without explanations).

A first observation is that the explanations have a positive effect on the *Utility-$K$* score: the explanation allows participants to better predict the model's decision (as the *Utility* scores are above 1). These results are consistent with those reported in [6]. This is confirmed with an Analysis of Variance (ANOVA) for which we found a significant main effect, with a medium effect size ($F(7, 234) = 9.19$, $p < .001$, $\eta^2 = 0.089$). Moreover, the only score below the baseline is that of the control explanation, which do not make use of the model. We further explore our results by performing pairwise comparisons using Tukey's Honestly Significant Difference [73] to compare the different explanations against the baseline. We found 3 explainability methods to be significantly better than the baseline: Grad-CAM ($p < 0.001$), Occlusion ($p = 0.01$) and SmoothGrad ($p = 0.034$). Thus, participants who received the Grad-CAM, Occlusion or SmoothGrad explanations performed much better than those who did not receive them.

---

[§]We acknowledge the existence of some overlap between the scenario 1 and scenario 3 as bias detection is a special case of a failure case. The reason we still use scenario 1 is because of the work previously done on it, allowing us to validate our framework.

**Scenario (2): Identifying an expert strategy**    In Fig. 3, we show results on the Leaves dataset. An ANOVA analysis across all conditions revealed a significant main effect, albeit small ($F(7, 232) = 4.29$, $p < .001$, $\eta^2 = 0.042$). This implies that explanation also had a positive effect resulting in better Meta-predictor in this use case. A Tukey's Honestly Significant Difference test suggests that the best explanations are Saliency, SmoothGrad and Integrated Gradients as they are the only ones to be significantly better than our baseline (WE) ($p = .004$, $p = .007$ and $p = .03$ respectively). An interesting result is that SmoothGrad seems to be consistently useful across both use cases where explanations are indeed practically useful. A more surprising result is that Saliency which was one of the worst explanations for bias detection, is now the best explanation on this use case (We discuss possible reasons in SI).

**Scenario (3): Understanding failure cases**    Table 1 shows that, on the ImageNet dataset, none of the methods tested exceeded baseline accuracy. Indeed, the experiment carried out, even with an improved experimental design, led us to the same conclusion as previous works [59]: none of the tested attribution methods are useful (ANOVA: $F(7, 233) = 1.26$, $p > .05$). In the use case of understanding failure cases on ImageNet, **no attributions methods** seem to be useful.

**Why do attribution methods fail?**

After studying the usefulness of attribution methods across 3 real-world scenarios for eXplainable AI, we found that attribution methods help, sometimes, but not always. We are interested in better understanding why sometimes attribution methods fail to help. Because this question has yet to be properly studied, there is no consensus if we can still make attribution methods work on those cases with incremental quantitative improvements. In the follow-up sections we explore 3 hypothesis to answer that question.

**Faithfulness as a proxy for Utility?**    Faithfulness is often described as one of the key desiderata for a good explanation [23, 74, 75]. If an explanation fails to be sufficiently faithful, the rules it highlights won't allow a user to understand the inner-working of the model. Thus, a lack of faithfulness on ImageNet could explain our results. To test this hypothesis, we use the *faithfulness* metrics: Deletion[9, 28], commonly used to compare attribution methods [9, 12, 24, 25, 28, 29]. A low Deletion score indicates a good *faithfulness*, thus for ease of reading we report the *faithfulness* score as $1-$ Deletion such that a higher *faithfulness* score is better.

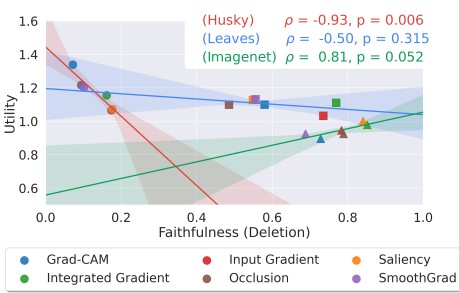

Figure 4: *Utility* **vs Faithfulness correlation.** The results suggest that current faithfulness metrics are poor predictors of the end-goal usefulness of explanation methods. Concerning the ImageNet dataset (triangle marker), the *Utility* scores are insignificant since none of the methods improves the baseline.

Fig 4 shows the linear relationship between our Utility metrics and the faithfulness scores computed for every attribution method across all 3 datasets. We observe two main trends: 1) **There does not appear to be any specific pattern regarding faithfulness that could explain why attribution methods are not useful for ImageNet**, and 2) the least useful attribution methods for both use cases for which methods help (Bias and Leaves) are some of the leading methods in the field measured by the faithfulness metric. We also found a weak, if maybe anti-correlated, relation between faithfulness and usefulness: just focusing on making attribution methods more faithful does not translate to having methods with higher practical usefulness for end-users. And, in fact, focusing too heavily on faithfulness seem to come at the expense of usefulness, resulting in explanations that are counter-intuitively less useful. This second observation may seems rather alarming for the field given that the faithfulness measure is one of the driving benchmarks.

**Are explanations too complex?**    Using the trade-off between completeness and understandability previously discussed in Section 3, we formulate another hypothesis: some explanations may be faithful but too complex and therefore cannot be understood by humans. In that view, an explanation with low complexity would tend to be more useful.

As a simple measure of the complexity of visual explanations, it would be ideal to be able to compute the Kolmogorov complexity [76] of each explanation. It was shown in previ-

ous work [77] to correlate well with human-derived ratings for the complexity of natural images [78, 79]. As suggested by [76, 80] we used a standard compression technique (JPEG) to approximate the Kolmogorov complexity. Fig. 5 shows the *Utility* vs complexity score of attribution methods for each dataset. For one of the datasets where attribution methods help, the results suggest the presence of a strong correlation between *usefulness* and *complexity*: the least complex method is the most useful to end-users. For the other datasets, the results are either not conclusive (Leaves), or are not relevant as methods are not useful (ImageNet).

Overall, **across datasets there is no significant difference in the complexity of explanations** that can explain why attribution methods do not help on ImageNet. This could be because the Kolmogorov Complexity does not perfectly reflect human visual complexity, or because this is not the key element to explain failure cases of attribution methods.

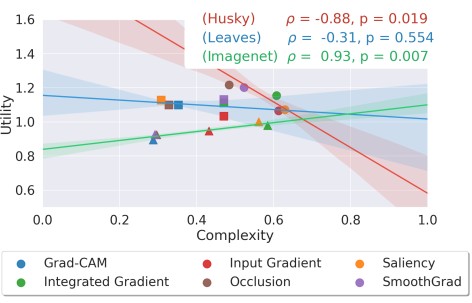

Figure 5: *Utility* **vs Complexity correlation.** This suggest a weak but possibly existing link between *Complexity* and *Utility* scores.

**An intrinsic limitation of Attribution methods?** The role of attribution methods is to help identify "where" to look in an image to understand the basis for a system's prediction. However, attribution methods do not tell us "what" in those image regions is driving decisions. For categorization problems which involve perceptually similar classes (such as when discriminating between different breeds of dogs) and fine-grained categorization problems more generally, simply looking at diagnostic image regions tells the user very little about the specific shape property being relevant. For instance, knowing that the ear shape is being used for recognition does not say what specific shape feature is being encoded (e.g., pointed vs. round or narrow vs. broad base, etc). Our main hypothesis is that such a lack of explicit "what" information is precisely what is driving the failure of attribution methods on our ImageNet use-case.

To test this hypothesis, we estimated the perceptual similarity between classes measured within diagnostic regions (see SI for more details) using the Learned Perceptual Image Patch Similarity (LPIPS) metric [81] as it has been shown to approximate human perceptual similarity judgments well [81, 82]. We report the perceptual similarity score as 1 - LPIPS score so that a high score means a high similarity. Fig. 6 shows the correlation between the perceptual similarity scores vs. our *Utility* scores on all methods and datasets studied. Our results suggest a strong correlation between perceptual similarity and practical usefulness: the more perceptually similar discriminative features of both classes are, the less useful attribution methods become. More importantly, the results across datasets show that on ImageNet, where attribution methods do not help, every method has a high similarity score. This result suggests that **after a certain threshold of perceptual similarity, attribution methods might no longer be useful, no matter how faithful or low in complexity the explanation**

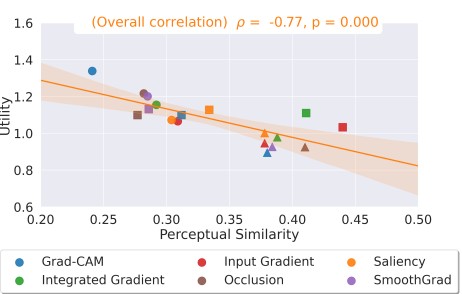

Figure 6: *Perceptual Similarity* **scores vs.** *Utility*. The perceptual similarity of highlighted regions by a given attribution method for both classes is measured, for each method, for each dataset. We observe a strong correlation between the perceptual similarity of features highlighted for both classes and the practical usefulness of methods.

**is.** Overall, the results suggest that the perceptual similarity of discriminative features could explain why attribution methods fail on ImageNet.

## 6    Discussion

In summary, we conducted a large-scale human psychophysics experiment to test the utility of explainability methods in real-world scenarios. Our work shows that in two of the three tested scenarios (bias detection and identification of new strategies), explainability methods have indeed

progressed and they provide meaningful assistance to human end-users. Nevertheless, we identified a scenario (understanding failure case) for which none of the tested attribution methods were helpful. This result is consistent with previous work [59] and highlights a fundamental challenge for XAI.

Further analysis of associated faithfulness performance metrics driving the development of explainability methods revealed that they did not correlate with our empirical measure of utility – suggesting that they might not be suited anymore to move the field forward. We also investigated the possibility that the complexity of individual explanations may play a role in explaining human failures to learn to leverage those explanations to understand the model and, while we found a weak correlation between complexity and our empirical measure of utility, this correlation appears too low to explain the failure of these methods.

Finally, because attribution methods appear to be just as faithful and low in complexity whether they are useful or not, we explored the possibility that their failure lies, not in the quality of their explanations, but in the intrinsic limitations of attribution methods. If fully grasping the strategy of a model requires understanding, not just "where" to look (as revealed by attribution maps) but also "what" to look at, something not currently revealed by these methods, attribution methods will not help. Our assumption is that the need for finer "what" information should arise when diagnostic image locations across classes look perceptually very similar and potentially semantically related for certain classification problems (e.g., looking at the ears or the snout to discriminate between breeds of cats and dogs) and one needs to identify what visual features are driving decisions. We computed a perceptual score for classification problems by estimating the perceptual similarity between diagnostic image regions (as predicted by attribution methods) and found that, indeed, when this score predicts a certain level of perceptual similarity between classes, attribution methods fail to contribute useful information to human users, regardless of the faithfulness or complexity of the explanations. This suggests that explainability methods may need to communicate additional information to the end user beyond attribution maps.

# 7   Limitation and broader impact

**Limitations.**   Our definition of the usefulness of an explanation is quite general. However, there are several limitations associated with our approach to estimate usefulness. First, our approach does not take into account the level of machine learning expertise of the user (the most useful explanation for a novice might not be the best one for an expert). Second, the need for human participants to evaluate explanations brings challenges compared with the automated metrics currently used in the field. However, by releasing all our data and software [¶], we hope to encourage the adoption of the approach to evaluate future explainability methods and to assess the overall progress towards the development of more human-interpretable AI systems.

**Broader impacts.**   While the increasing use of AI in real-world scenarios has shown potential to do good [46–48], it has also shown its potential for harm, especially when models rely on shortcuts (e.g., relying on spurious correlations in the training set that leads to unintended racial bias [39–41]). To identify such shortcuts, the field of Explainable AI has developed a lot of explainability methods; but it is not always clear which one performs best when trying to understand a model. We hope our evaluation method can help in this regard, assisting the AI practitioner in better identifying bias and shortcuts that may unfairly discriminate against groups of people.

## Acknowledgments

This work was conducted as part the DEEL[‖] project. It was funded by ONR grant (N00014-19-1-2029), NSF grant (IIS-1912280 and EAR-1925481), and the Artificial and Natural Intelligence Toulouse Institute (ANITI) grant #ANR19-PI3A-0004. The computing hardware was supported in part by NIH Office of the Director grant #S10OD025181 via the Center for Computation and Visualization (CCV) at Brown University. J.C. has been partially supported by funding from the Valencian Government (Conselleria d'Innovació, Universitats, Ciència i Societat Digital) by virtue of a 2022 grant agreement (convenio singular 2022).

---

[¶]github.com/serre-lab/Meta-predictor

[‖]https://www.deel.ai/

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
