# A  Human experiments

## A.1  Experimental design

Figure 1 summarizes the experimental design used for our experiments. The participants that went through our experiments are users from the online platform Amazon Mechanical Turk (AMT). Through this platform, users stay anonymous, hence, we do not collect any sensitive personal information about them. We prioritized users with a Master qualification (which is a qualification attributed by AMT to users who have proven to be of excellent quality) or normal users with high qualifications (number of HIT completed = 10000 and HIT accepted > 98%).

Before going through the experiment, participants are asked to read and agree to a consent form, which specifies: the objective and procedure of the experiment, as well as the time expected to completion ($\sim$ 5 - 8 min) with the reward associated ($1.4), and finally, the risk, benefits, and confidentiality of taking part in this study. There are no anticipated risks and no direct benefits for the participants taking part in this study.

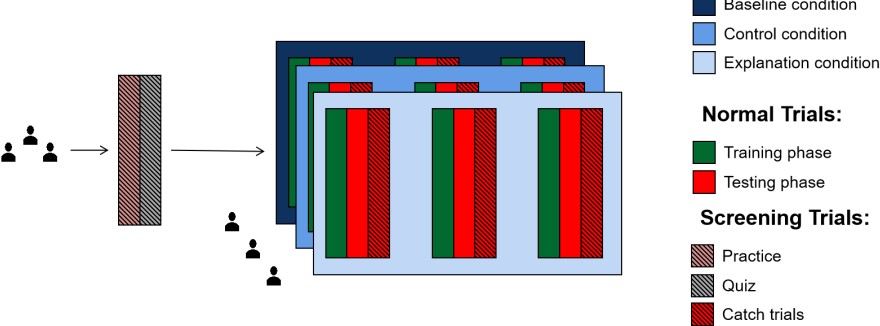

Figure 1: **Experimental design.** First, every participant goes through a practice session (fig 2) to make sure they understand how to use attribution methods to infer the rules used by a model, and a quiz (fig 3) to make sure they actually read and understand the instructions. Then, participants are split into the different conditions – every participant will only go through one condition. The 3 possible conditions are: an Explanation condition where an explanation is provided to human participants during their training phase, a Baseline condition where no explanation was provided to the human participants, and a Control condition where a non-informative explanation was provided. The main experiment was divided into 3 training sessions each followed by a brief test. In each individual training trial, an image was presented with the associated prediction of the model, either alone for the baseline condition or together with an explanation for the experimental and control condition. After a brief training phase (5 samples), participants' ability to predict the classifier's output was evaluated on 7 new samples (only the image, no explanation) during a test phase. To filter out uncooperative participants we also add a catch trial (fig 4) in each test session.

**Controlling for prior class knowledge**  To control for users' own semantic knowledge, we balanced the samples shown to participants so that the classifiers were correct/incorrect 50% of the time. This way, the baseline (participants who try to simply predict the true class label of an image as opposed to learning to predict the model's outputs) is at 50%. Any higher score reflects a certain understanding of the rules used by the model.

## A.2  Pruning out uncooperative participants

**3-stage screening proccess.**  To prune out uncooperative participants, we subjected them to a 3-stage screening process. First, participants completed a short practice session to make sure they understood the task and how to use the attribution methods to infer the rules used by the model (fig 2). Second, as done in [1], we asked participants to answer a few questions regarding the instructions provided to make sure they actually read and understood them (fig 3). Third, during the main experiment, we took advantage of the reservoir to introduce a catch trial (fig 4). The reservoir is the place where we store the training example of the current session, which can be accessed during the testing phase. We added a trial in the testing phase of each session where the input image corresponded to one of

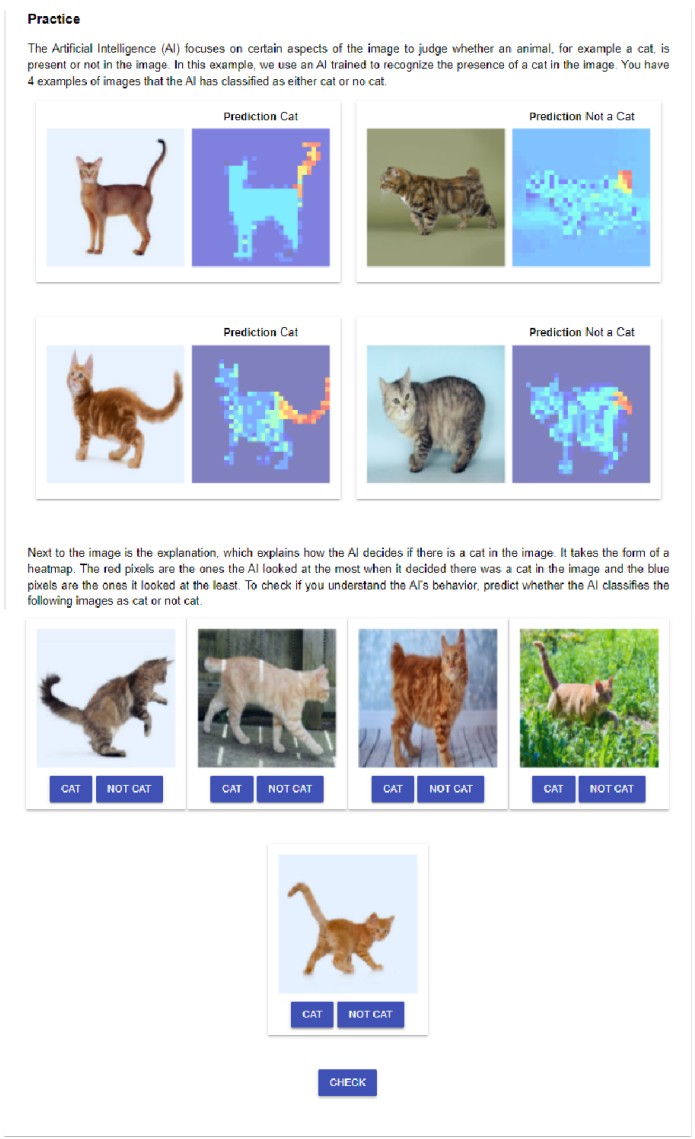

Figure 2: **Practice session.** Through a practice session, which is a simplified version of the main experiment, we evaluate if users understand how to read and use explanations. Participants that failed to predict correctly any of the 5 cat test images on the first try were excluded from further analysis.

the training samples used in the current session: since the answer is still on the screen (or a scroll away) we expect participants to be correct on these catch trials. Participants that failed any of the 3 screening processes were excluded from further analysis.

Practice session is now complete.

You will now begin the main experiment.
All the images of Red fox and Kit fox that will be shown in this experiment were also shown to an AI that was trained to recognize Red fox and Kit fox in pictures.
Unfortunately, our AI is not perfect, it is not always able to predict the right animal in the picture. For every picture, the AI was asked to guess whether there was a Red fox or a Kit fox in the picture. In total, it predicted Red fox half the time and the other half it predicted Kit fox.

**We hope you can help us understand the behavior of the AI.**

Your task will first be to study examples of model classification, to try to understand the AI's behavior: what makes it predict Red fox vs. Kit fox. Then you will make predictions on new images, which means you will have to put yourself in the shoes of the AI based on the behavior you have captured.
Concerning these sessions, there will be 3 of them, each one composed of a training phase (viewing 5 images, their predictions and an explanation if you are in the condition with explanation) and a testing phase (on 7 images, predict the AI's prediction)
During the testing phase, you will have access to the examples you studied previously (only the last 5 so as not to flood your screen with too many images).

To make sure instructions were clear, please answer the following question, thank you.

Does the AI always make the correct prediction ?
◉ Yes
○ No

Will you have access to the training examples as references while making prediction on new test images ?
◉ Yes
○ No

How many block (learning + testing) is there in the experiment?
◉ 1
○ 2
○ 3

Figure 3: **Quiz.** Through a quiz, we make sure that users read and understood the instructions. Participants that did not answer correctly every question on the first try were excluded from further analysis.

## A.3  More results

**Reaction time.** We explored whether the usefulness of a method is reflected in the reaction time of participants -i.e., the more useful the explanation the faster the participants are able to grasp the strategy of the model-. Table 1 shows the reaction time of participants across methods, across datasets. We do not find any trend linking reaction time with usefulness.

| Method | Husky vs. Wolf | Leaves | ImageNet |
|---|---|---|---|
| Saliency [2] | 207.7 | **212.9** | 202.3 |
| Integ.-Grad. [3] | 213.1 | 216.5 | 218.5 |
| SmoothGrad [4] | 215.8 | **268.8** | 243.9 |
| GradCAM [5] | **168.9** | 154.6 | 268.9 |
| Occlusion [6] | 221.2 | 229.2 | 274.4 |
| Grad.-Input [7] | 210.4 | 238.1 | 208.0 |

Table 1: **Average total *time* per method per dataset (in second).** For each dataset, we **bold** the most useful method, and we underline the least useful method.

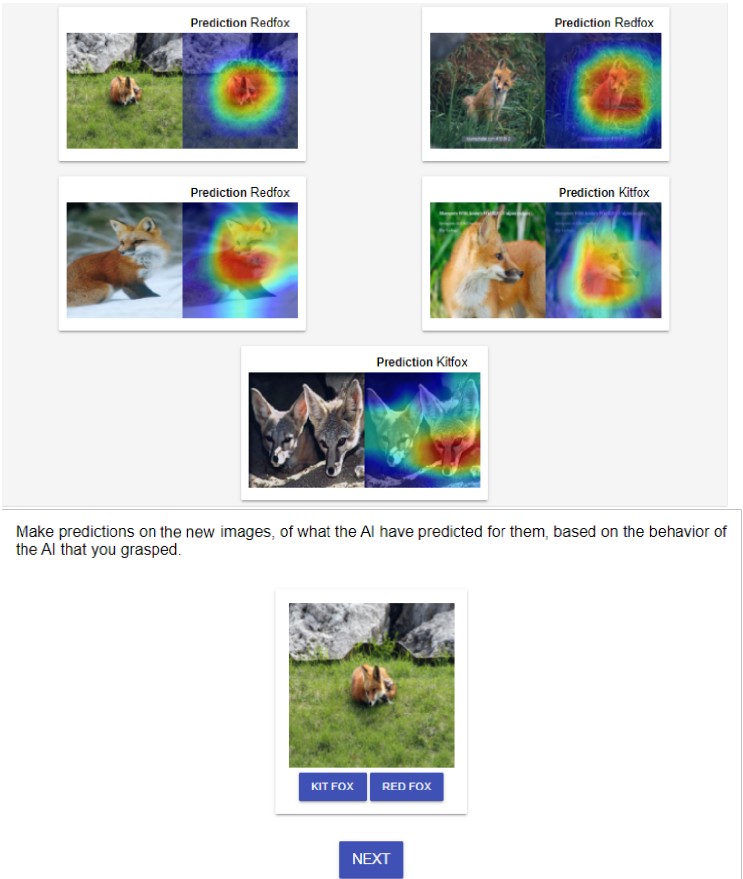

Figure 4: **Catch trial.** We use a reservoir (to store all the examples of the current training session) that participants can refer to during the testing phase to minimize memory load. At the top of the screen is the reservoir, at the bottom of the screen is a trial from the testing phase. We take advantage of the reservoir to introduce a catch trial. We added a trial in the testing phase of each session where the input image corresponded to one of the training samples used in the current session: since the answer is still on the screen (or a scroll away) we expect participants to be correct on these catch trials. Participants that failed any of the 3 catch trials (one per session) were excluded from further analysis.

## B  Why do the best methods for the use cases Bias detection and Identifying an expert strategy (leaves) differ?

The most interesting case is Saliency, which is the worst method on the bias dataset but the best on the "leaves" dataset. On the bias dataset, the model seems to focus on the background (i.e., a coarse feature), and on the "leaves" dataset the model seems to focus either on the margin or on the vein of the leaf (i.e., very fine features). We hypothesize that different methods suit different granularity of features (coarse vs fine). [4] make the hypothesis that "the saliency maps are faithful descriptions of what the network is doing" but because "the derivative of the score function with respect to the input [is] not [...] continuously differentiable", the saliency map can appear noisy. Because of this local discontinuity of the gradient, a large patch of important pixels is often portrayed in the saliency map as a collection of smaller patches of important pixels (i.e., a coarse feature vs multiple individual fine features) which can make it hard to identify if the strategy is the coarse feature or a more complex interaction of the smaller features. In the bias dataset, because the model relies on the background, the Saliency maps appear very noisy and the explanation ends-up not being useful. We note that SmoothGrad, which proposes to fix that discontinuity, is useful. On the other hand, on the leaves dataset, the model uses very fine features, therefore the Saliency maps suffer less from the discontinuity, it does not appear noisy, Saliency is useful. We also note that in this case, SmoothGrad is not better than Saliency, which can arguably be attributed to the fact that we do not need to fix the discontinuity of the gradient. Conversely, because the granularity of both Grad-CAM (the feature map is much smaller than image size) and Occlusion (the patch size is much bigger than a pixel) is too high, the heatmaps they offer on the "leaves" dataset are too coarse to specifically highlight the fine features and it seems to take more time for the subjects to pick-up on them. But on the biased dataset, Grad-CAM and Occlusion are the best performing methods.

## C  Why do attribution methods fail?

### C.1  Faithfulness

While the Deletion[8] measure is the most commonly used faithfulness metric, for completeness we also consider 2 others faithfulness metric available in the Xplique library[9]: Insertion[8] and $\mu$Fidelity[10]. Fig 5 shows the correlation between either measure and our *Utility*. We find them to be no better predictor of the practical usefulness of attribution methods than the Deletion measure.

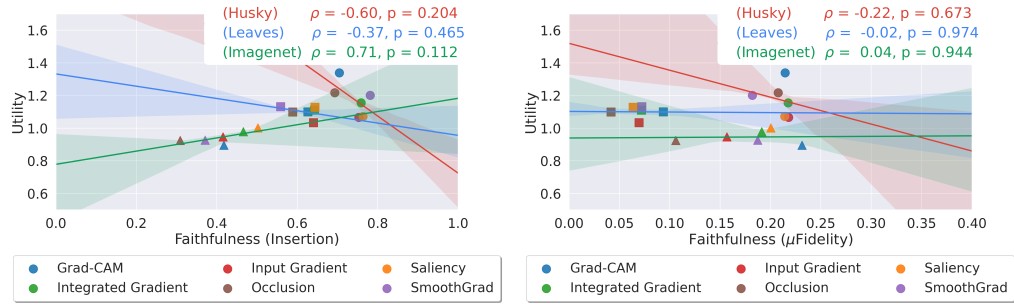

Figure 5: *Utility* **vs Insertion correlation &** *Utility* **vs** $\mu$**Fidelity correlation** The results suggest that every faithfulness metrics tested are poor predictors of the practical usefulness of attribution methods. Concerning the ImageNet dataset (triangle marker), the *Utility* scores are insignificant since none of the methods improves the baseline.

### C.2  Perceptual Similarity

Tab 2 shows the Perceptual Similarity scores obtained for each method, on every dataset. We observe that on ImageNet, where attribution methods do not help, the perceptual similarity scores are clearly higher than on the two other datasets, where attribution methods help.
Fig 6 shows examples of patches for each dataset using Grad-CAM.

| Method | Husky vs. Wolf | Leaves | ImageNet |
|---|---|---|---|
| Saliency [2] | 0.304 | 0.334 | **0.378** |
| Integ.-Grad. [3] | 0.292 | **0.411** | **0.388** |
| SmoothGrad [4] | 0.285 | 0.286 | **0.384** |
| GradCAM [5] | 0.241 | 0.312 | **0.38** |
| Occlusion [6] | 0.282 | 0.277 | **0.41** |
| Grad.-Input [7] | 0.309 | **0.44** | **0.378** |

Table 2: *Perceptual Similarity* **scores.** The perceptual similarity of highlighted regions by a given attribution method for both classes is measured, for each method, for each dataset. The perceptual similarity scores that are higher than $0.378$ (the minimum score on ImageNet) are **bolded**. Higher is more similar.

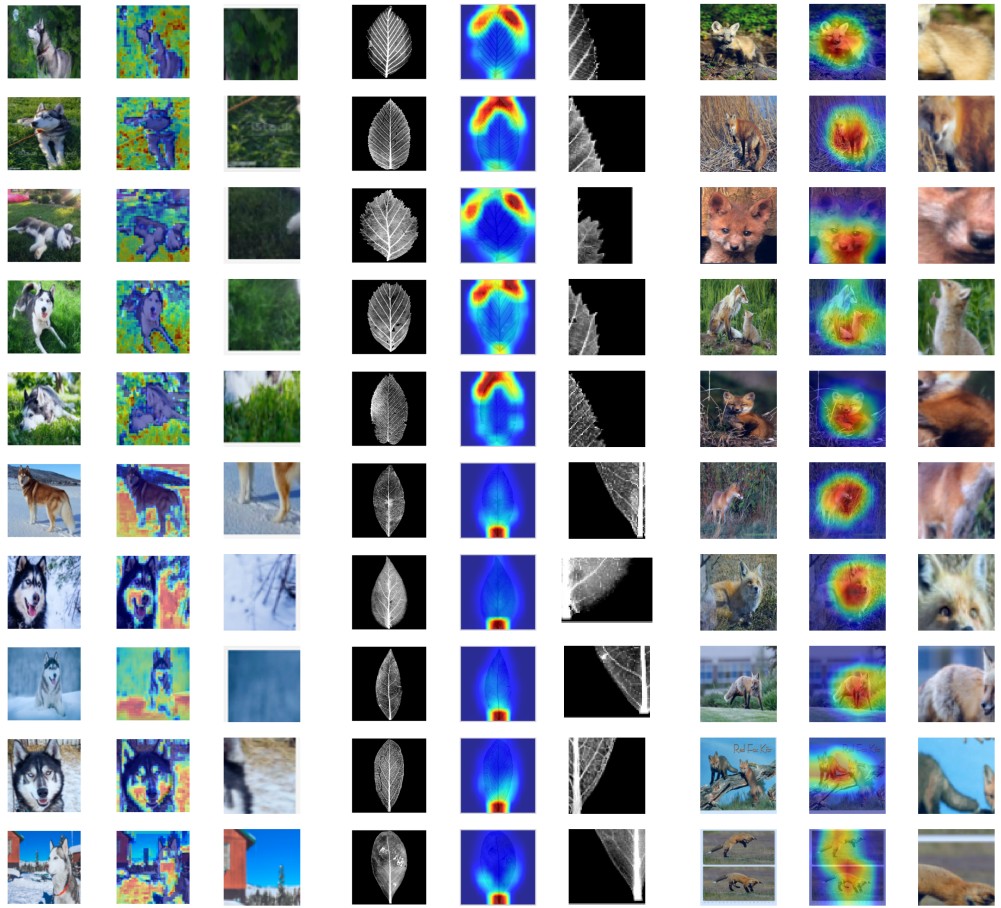

Figure 6: **Examples of extracted patches.** The perceptual similarity score is performed on the locations considered most important by the attribution methods. Examples of patches extracted for the three datasets with the Grad-CAM method.

# D Attribution methods

## D.1 Methods

In the following section, the formulation of the different methods used in the experiment is given. We define $f(x)$ the logit score (before softmax) for the class of interest. An explanation method provides an attribution score for each input variables. Each value then corresponds to the importance of this feature for the model results.

**Saliency** [2] is a visualization technique based on the gradient of a class score relative to the input, indicating in an infinitesimal neighborhood, which pixels must be modified to most affect the score of the class of interest.

$$\Phi^{SA}(x) = ||\nabla_x f(x)||$$

**Gradient $\odot$ Input** [7] is based on the gradient of a class score relative to the input, element-wise with the input, it was introduced to improve the sharpness of the attribution maps. A theoretical analysis conducted by [11] showed that Gradient $\odot$ Input is equivalent to $\epsilon$-LRP and DeepLIFT [12] methods under certain conditions: using a baseline of zero, and with all biases to zero.

$$\Phi^{GI}(x) = x \odot ||\nabla_x f(x)||$$

**Integrated Gradients** [3] consists of summing the gradient values along the path from a baseline state to the current value. The baseline is defined by the user and often chosen to be zero. This integral can be approximated with a set of $m$ points at regular intervals between the baseline and the point of interest. In order to approximate from a finite number of steps, we use a Trapezoidal rule and not a left-Riemann summation, which allows for more accurate results and improved performance (see [13] for a comparison). The final result depends on both the choice of the baseline $\mathbf{x}_0$ and the number of points to estimate the integral. In the context of these experiments, we use zero as the baseline and $m = 80$.

$$\Phi^{IG}(x) = (x - x_0) \int_0^1 \nabla_x f(x_0 + \alpha(x - x_0)))d\alpha$$

**SmoothGrad** [4] is also a gradient-based explanation method, which, as the name suggests, averages the gradient at several points corresponding to small perturbations (drawn i.i.d from a normal distribution of standard deviation $\sigma$) around the point of interest. The smoothing effect induced by the average helps reduce the visual noise and hence improve the explanations. In practice, Smoothgrad is obtained by averaging after sampling $m$ points. In the context of these experiments, we took $m = 80$ and $\sigma = 0.2$ as suggested in the original paper.

$$\Phi^{SG}(x) = \mathbb{E}_{\varepsilon \sim \mathcal{N}(0, I\sigma)}(\nabla_x f(x + \varepsilon))$$

**Grad-CAM** [5] can be used on Convolutional Neural Network (CNN), it uses the gradient and the feature maps $\mathbf{A}^k$ of the last convolution layer. More precisely, to obtain the localization map for a class, we need to compute the weights $\alpha_c^k$ associated to each of the feature map activation $\mathbf{A}^k$, with $k$ the number of filters and $Z$ the number of features in each feature map, with $\alpha_k^c = \frac{1}{Z} \sum_i \sum_j \frac{\partial f(x)}{\partial \mathbf{A}_{ij}^k}$ and

$$\Phi^{GC} = max(0, \sum_k \alpha_k^c \mathbf{A}^k)$$

Notice that the size of the explanation depends on the size (width, height) of the last feature map, a bilinear interpolation is performed in order to find the same dimensions as the input.

**Occlusion** [6] is a sensitivity method that sweeps a patch that occludes pixels over the images, and uses the variations of the model prediction to deduce critical areas. In the context of these experiments, we took a patch size and a patch stride of of 1 tenth of the image size.

$$\Phi_i^{OC} = f(x) - f(x_{[x_i=0]})$$

## D.2 Examples of explanations

Examples of explanations from the different attributions methods evaluated through our experiments on the Husky vs. Wolf dataset (fig 7), the Leaves dataset (fig 8) and the ImageNet dataset (fig 9).

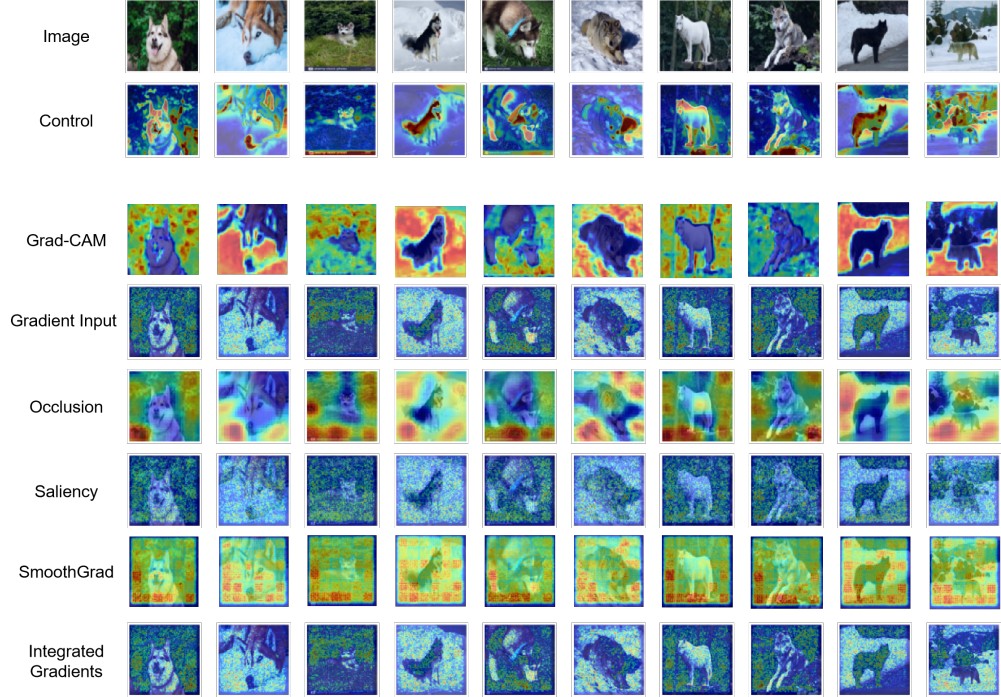

Figure 7: Examples of images from the Wolf vs. Husky experiment, alongside their respective: Control explanation (which is a non-informative explanation) as well as the different Attribution methods evaluated in our experiment.

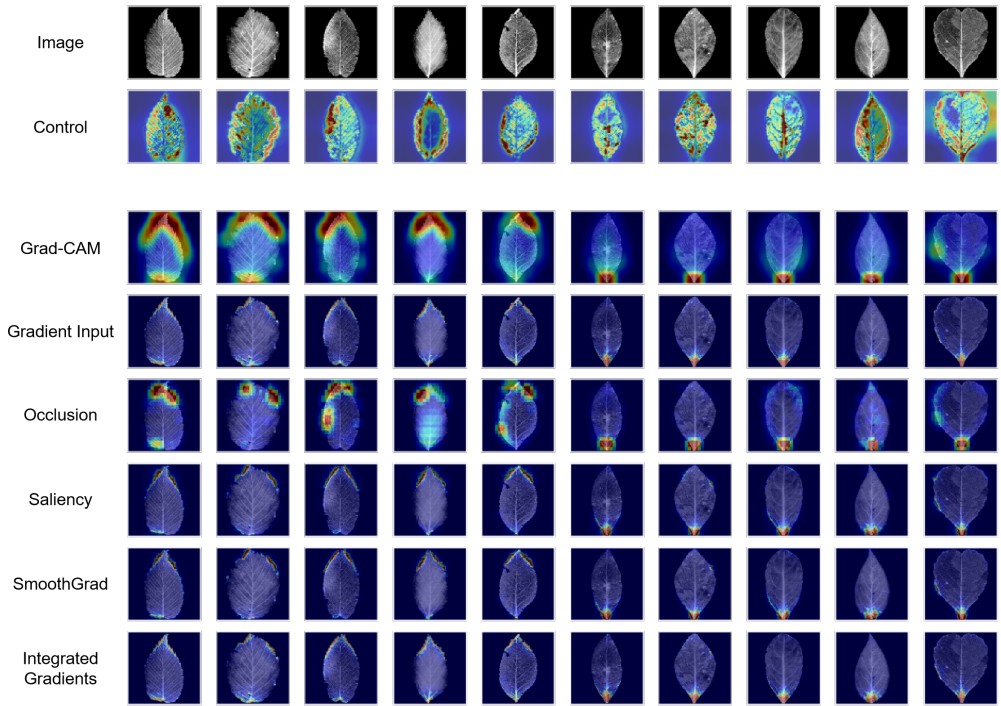

Figure 8: Examples of images from the Leaves experiment, alongside their respective: Control explanation (which is a non-informative explanation) as well as the different Attribution methods evaluated in our experiment.

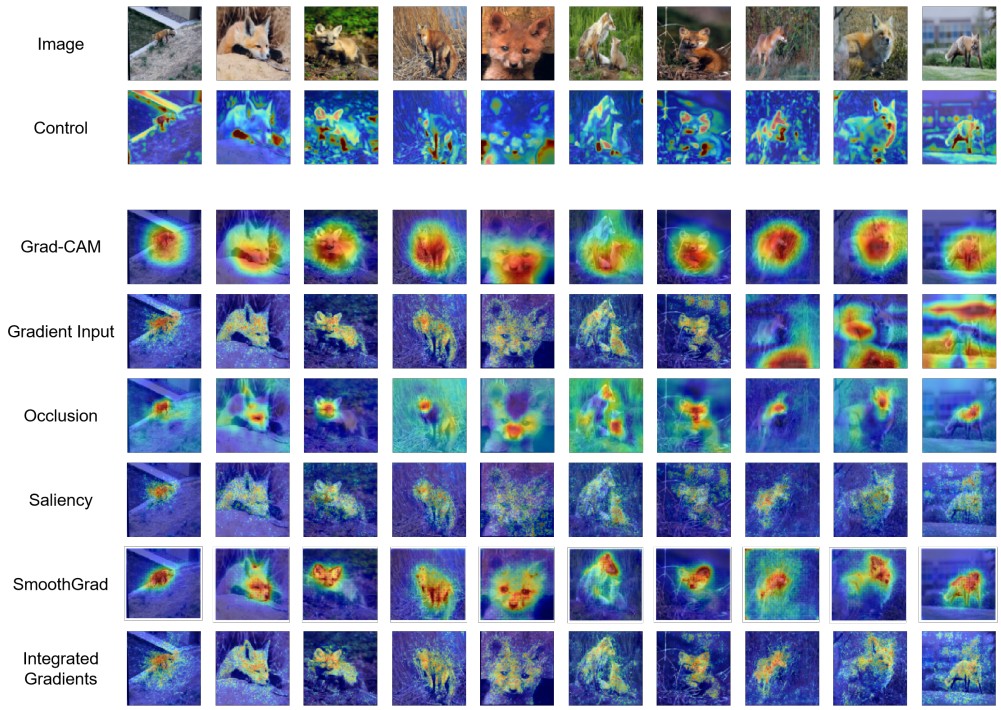

Figure 9: Examples of images from the ImageNet experiment, alongside their respective: Control explanation (which is a non-informative explanation) as well as the different Attribution methods evaluated in our experiment.