# OpenReview forum: "What I Cannot Predict, I Do Not Understand: A Human-Centered Evaluation Framework for Explainability Methods"
_NeurIPS.cc/2022/Conference — NeurIPS 2022 Accept_

### Official Review · Reviewer_LEiC · 2022-07-05

**Rating:** 5
**Confidence:** 4
**Soundness:** 3 good
**Presentation:** 3 good
**Contribution:** 2 fair

**Summary:**

The authors present a human-centered framework to evaluate the usefulness of attribution explainability methods. They evaluate 6 different attribution methods across 3 setttings. They show that generally, perceptual scores derived from attribution maps can be more useful at predicting failure cases of explainability methods better than fidelity metrics.

**Questions:**

In general I recommend rebranding the paper as a study of attribution methods instead of a framework, since the results are hard to scale, hard to reproduce and it is cumbersome to "integrate" a new explainer into this framework. However, the manuscript as is could be improved in the following ways:

#### Critical changes:
 - Provide some experiments with a varying number of classes or at least a small section discussing how the effectiveness of the meta-predictor approach varies when increasing the number of classes.

- The main reason the authors provide for the failure of attribution methods is an intrinsic limitation of said methods, since they can only show what regions in an image should be changed but not how. This is precisely one of  the limitiations counterfactual explainers solve, and since the authors claim that the framework can be applied to any explainer it would be useful to explore if counterfactual explainers fail in the same way, whilst substantiating said claim.
- It seems to me that there is a potential overlap within the 3 scenarios used for evaluation (Bias detection, Identifying an expert strategy, Understanding failure cases). For example: In scenario 2 (Identifying an expert strategy) whenever a meta-predictor is provided with an image that the model has failed to predict, then the task is no longer: _to understand the strategy that was discovered by the AI system_, but instead it becomes either bias detection (scenario 1) or Understanding failure cases (scenario 3). Conversely, in scenario 3 (if the classes are really similar) whenever the meta-predictor is provided with an image that the model has successfully predicted then the task becomes Identifying an expert strategy (scenario 2). An easy fix would be to provide only correctly and incorrectly classified samples for scenarios 2 and 3 respectively, however as stated by the authors this would significantly affect the meta-predictor score if the meta-predictor has some prior knowlege about the domain (dataset) the scenario is being evaluated on. If this overlap does in fact exists, I think it should be discussed in the paper.

#### Minor changes:

- Figure 1 is not referenced anywhere, also there is a missing reference on line 320
- In the section describing the understandability-completeness trade-off it would useful to mention/show examples of how an _entirely faithful_ and _more understandable_ explanations look like.
- In the General study design section it is mentioned that each training session has 5 training samples, but the plots showcasing the results show 5, 10 and 15, this should be clarified.
- I think it would be useful to incorporate the meta-predictor response time into the metric since an explanation that is correctly predicted instantly is probably better than one that gets correctly predicted after some time. Conversely an explanation that is incorrectly predicted instantly could be considered more "misleading" than one that gets incorrectly predicted after some time.

#### Questions:
- Do the authors have any hypothesis as to why the best method for Bias detection and Identifying an expert strategy differ? I feel like the paper would be more complete with a small section explaining this phenomenon, not unlike the section: __Why do attribution methods fail?__. In which the authors investigate why all methods underperform on the 3rd setting.

**Limitations:**

The limitations of the work should be discussed beyond the cost of human evaluation, since the framework itself has some limitations (mentioned above) such as scalability and reproducibility.

**Strengths And Weaknesses:**

Strengths:
- The explainability evaluation framework the authors propose:
   - is human-centered, which is, ideally, how the usefulness of explanations should be evaluated.
   - includes a control group, which was lacking in previous work

-  The paper is easy to understand and generally well written.
- The proposed metric is original and sound to the best of my knowledge.

Weaknesses:
- The proposed framework poses some reproducitiblity concerns since in order to evaluate any explainability one would have to undergo the entire process described in the paper of hiring and filtering the meta-predictors. This also makes it hard to scale.

- I worry about the effectiveness of the meta-predictor whenever evaluating a dataset with more than 2 classes.

- The authors state that the framework can be applied to other explainability methods and while this in principle makes sense, there are no experimental results supporting this claim.

---

> ### Author Response · Authors · 2022-08-02
> **Response to reviewer LEiC (Part I)**
>
> Thank you for your thorough review, thoughtful comments and helpful suggestions! Please kindly find our replies below.
>
> **The proposed framework poses some reproducitiblity concerns since in order to evaluate any explainability one would have to undergo the entire process described in the paper of hiring and filtering the meta-predictors. This also makes it hard to scale.**
>
> >This comment is misguided. A reproducibility concern would arise if the methods were not described in sufficient detail to allow someone else to run the same exact set of experiments. This is not the case here since we will provide the code and all necessary details. Filtering participants for online experiments is standard and necessary to remove participants that are not properly engaged or following explanations [1,2]. We agree that the use of human participants creates a burden on the researcher over automated scores but given that our results suggest that current scores are sometimes misleading we do not see any alternatives at the moment. We would like to highlight that this framework is one of the very few human studies [3] that can be scaled, as all code will be made available and all types of explainability methods can be tested.
>
> **I worry about the effectiveness of the meta-predictor whenever evaluating a dataset with more than 2 classes.**
> **Provide some experiments with a varying number of classes or at least a small section discussing how the effectiveness of the meta-predictor approach varies when increasing the number of classes**
>
> >Having to classify more than 2 classes would increase the participants’s cognitive load and bring unnecessary difficulty to the task.  This is not necessary because classification problems with more than 2 classes can always be reformulated as one vs. all / binary classification problems without lack of generality. Our task could be trivially reformulated as predicting Target class vs Other, instead of Class 1 vs Class 2 as we did in the paper. We will integrate a discussion about this in the final version of the manuscript.
>
> **The authors state that the framework can be applied to other explainability methods and while this in principle makes sense, there are no experimental results supporting this claim.**
>
> >While it is true that the paper focuses on attribution methods because they are the main method used in the field and deserve a proper study, we give a formal definition of the framework L.131 which explains why this framework can, and will, be used in future works for other explainability methods.
>
> **In general I recommend rebranding the paper as a study of attribution methods instead of a framework, since the results are hard to scale, hard to reproduce and it is cumbersome to "integrate" a new explainer into this framework.**
>
> >We thank the reviewer for offering valuable suggestions to improve the paper, but regarding this specific suggestion, we respectfully disagree. We will empathize this point in the final version, but the paper is both about a framework as well as an instantiation of this framework to study the pitfalls of attribution methods.
> We hope to have addressed the reviewer’ concerns about scaling and reproducibility. Regarding the last concern, we argue that the benefit for the community to have common human-centered evaluations outweighs the challenge associated with running human experiments.
>
> **The main reason the authors provide for the failure of attribution methods is an intrinsic limitation of said methods, since they can only show what regions in an image should be changed but not how. This is precisely one of the limitiations counterfactual explainers solve, and since the authors claim that the framework can be applied to any explainer it would be useful to explore if counterfactual explainers fail in the same way, whilst substantiating said claim.**
>
> >We are also interested in evaluating counterfactuals on the use case where attribution methods are not useful. But this would constitute future work because our main focus is to show that attribution methods do not help because they lack the “what” component, which is independent of whether or not current counterfactual explainers address this concern.
>
> **In the General study design section it is mentioned that each training session has 5 training samples, but the plots showcasing the results show 5, 10 and 15, this should be clarified.**
>
> >The plot showcases the total number of training samples seen at that point of the testing phase. During the 1rst testing phase, they’ve seen 5 samples from the 1rst training phase, during the 2nd testing phase, they’ve seen in total, 5 from 1rst training phase + 5 from 2nd training phase = 10 samples, etc ... We will clarify that in the legend of the figure.

---

> > ### Author Response · Authors · 2022-08-02
> > **Response to reviewer LEiC (Part II)**
> >
> > **In the section describing the understandability-completeness trade-off it would useful to mention/show examples of how an entirely faithful and more understandable explanations look like.**
> >
> > >Thank you for your suggestion, we will add examples of those explanations in the final version of the manuscript!
> >
> > **It seems to me that there is a potential overlap within the 3 scenarios used for evaluation (Bias detection, Identifying an expert strategy, Understanding failure cases). For example: In scenario 2 (Identifying an expert strategy) whenever a meta-predictor is provided with an image that the model has failed to predict, then the task is no longer: to understand the strategy that was discovered by the AI system, but instead it becomes either bias detection (scenario 1) or Understanding failure cases (scenario 3). Conversely, in scenario 3 (if the classes are really similar) whenever the meta-predictor is provided with an image that the model has successfully predicted then the task becomes Identifying an expert strategy (scenario 2). An easy fix would be to provide only correctly and incorrectly classified samples for scenarios 2 and 3 respectively, however as stated by the authors this would significantly affect the meta-predictor score if the meta-predictor has some prior knowlege about the domain (dataset) the scenario is being evaluated on. If this overlap does in fact exists, I think it should be discussed in the paper.**
> >
> > >We agree that the problem of overlap and distinction between use cases of XAI in general is not always clear, but important, and we will add a discussion about it in the paper. Here specifically, we agree that there exists some overlap between the scenario 1 and 3, as bias detection is a special case of a failure case (we use scenario 1 because it was already used in the literature before and it allows us to validate that our framework works). Regarding scenario 2, we only provide correctly classified samples as pointed out by the reviewer (and model accuracy on those classes is 90.2%), hence we think it is reasonable to say that this is a scenario of identifying an expert strategy.
> >
> > **Do the authors have any hypothesis as to why the best method for Bias detection and Identifying an expert strategy differ? I feel like the paper would be more complete with a small section explaining this phenomenon, not unlike the section: Why do attribution methods fail?. In which the authors investigate why all methods underperform on the 3rd setting.**
> >
> > >This is a very good question! The most interesting case is Saliency, which is the worst method on the bias dataset but the best on the “leaves” dataset. On the bias dataset, the model seems to focus on the background (i.e., a coarse feature), and on the “leaves'' dataset the model seems to focus either on the margin or on the vein of the leaf (i.e., very fine features). We hypothesize that different methods suit different granularity of features (coarse vs fine). [4] make the hypothesis that “the saliency maps are faithful descriptions of what the network is doing” but because “the derivative of the score function with respect to the input [is] not [...] continuously differentiable”, the saliency map can appear noisy. Because of this local discontinuity of the gradient, a large patch of important pixels is often portrayed in the saliency map as a collection of smaller patches of important pixels (i.e., a coarse feature vs multiple individual fine features) which can make it hard to identify if the strategy is the coarse feature or a more complex interaction of the smaller features. In the bias dataset, because the model relies on the background, the Saliency maps appear very noisy and the explanation ends-up not being useful. We note that SmoothGrad, which proposes to fix that discontinuity, is useful. On the other hand, on the leaves dataset, the model uses very fine features, therefore the Saliency maps suffer less from the discontinuity, it does not appear noisy, Saliency is useful. We also note that in this case, SmoothGrad is not better than Saliency, which can arguably be attributed to the fact that we do not need to fix the discontinuity of the gradient. Conversely, because the granularity of both Grad-CAM (the feature map is much smaller than image size) and Occlusion (the patch size is much bigger than a pixel) is too high, the heatmaps they offer on the “leaves” dataset are too coarse to specifically highlight the fine features and it seems to take more time for the subjects to pick-up on them. But on the biased dataset, Grad-CAM and Occlusion are the best performing methods. We will add this discussion to the final version of the manuscript.

---

> > > ### Author Response · Authors · 2022-08-02
> > > **Response to reviewer LEiC (Part III)**
> > >
> > > **I think it would be useful to incorporate the meta-predictor response time into the metric since an explanation that is correctly predicted instantly is probably better than one that gets correctly predicted after some time. Conversely an explanation that is incorrectly predicted instantly could be considered more "misleading" than one that gets incorrectly predicted after some time.**
> > >
> > > >We agree with the reviewer that response time can be an interesting complement to the main metric to decide between 2 equally informative methods. We will add a discussion about the response time in the SI for the final version of the manuscript.
> > >
> > > **The limitations of the work should be discussed beyond the cost of human evaluation, since the framework itself has some limitations (mentioned above) such as scalability and reproducibility.**
> > >
> > > >We have incorporated a section about limitations and societal impact in SI.
> > >
> > > * [1] Oppenheimer, Daniel M., Tom Meyvis, and Nicolas Davidenko. "Instructional manipulation checks: Detecting satisficing to increase statistical power." Journal of experimental social psychology 45.4 (2009): 867-872.
> > > * [2] Berinsky, Adam J., Michele F. Margolis, and Michael W. Sances. "Separating the shirkers from the workers? Making sure respondents pay attention on self‐administered surveys." American Journal of Political Science 58.3 (2014): 739-753.
> > > * [3] Kim, S. S., Meister, N., Ramaswamy, V. V., Fong, R., & Russakovsky, O. “HIVE: Evaluating the Human Interpretability of Visual Explanations”. In: European Conference of Computer Vision (ECCV). 2022.
> > > * [4] Smilkov, D., Thorat, N., Kim, B., Viégas, F., & Wattenberg, M. (2017). Smoothgrad: removing noise by adding noise. Arxiv 2017

---

> > > ### Comment · Reviewer_LEiC · 2022-08-06
> > > **Last Concern**
> > >
> > > __We agree that the problem of overlap and distinction between use cases of XAI in general is not always clear, but important, and we will add a discussion about it in the paper. Here specifically, we agree that there exists some overlap between the scenario 1 and 3, as bias detection is a special case of a failure case (we use scenario 1 because it was already used in the literature before and it allows us to validate that our framework works). Regarding scenario 2, we only provide correctly classified samples as pointed out by the reviewer (and model accuracy on those classes is 90.2%), hence we think it is reasonable to say that this is a scenario of identifying an expert strategy.__
> > >
> > > >Since for scenario 2 you only provide correctly classified samples, I assume that for scenario 3 you only provide incorrectly classified samples (otherwise there would be an overlap with scenario 2) is this assumption correct? Further, this should be clarified in the paper because right now  the authors mention a balance ratio of correctly and incorrectly classified samples when describing scenario 1 ("_In this situation where prior knowledge of subjects can affect their Meta-predictor score, we balance data correctness (50% of correct/incorrect examples shown"_) but make no such clarification for scenario 2 and 3.
> > >
> > > __This is a very good question! The most interesting case is Saliency,..__
> > >
> > > >Thanks for such a detailed answer!

---

> > > > ### Author Response · Authors · 2022-08-09
> > > > **RE: Last Concern**
> > > >
> > > > **Since for scenario 2 you only provide correctly classified samples, I assume that for scenario 3 you only provide incorrectly classified samples (otherwise there would be an overlap with scenario 2) is this assumption correct? Further, this should be clarified in the paper because right now since the authors mention a balance ratio of correctly and incorrectly classified samples when describing scenario 1 ("In this situation where prior knowledge of subjects can affect their Meta-predictor score, we balance data correctness (50% of correct/incorrect examples shown") but make no such clarification for scenario 2 and 3.**
> > > >
> > > > >To be clear: for scenario 3, while providing only incorrectly classified samples makes perfect sense, we provide 50% of correct/incorrect examples because we have to control for participants' prior knowledge which would otherwise confound the results. But this does not mean that scenarios 2 and 3 overlap because we define “Identifying an expert strategy” as a case where participants do not already come equipped with the right strategy to classify, and they must learn the one successfully used by the model. In case 3, the participants already come equipped with the right strategy, therefore, this assumption does not hold. We mention the balance ratio, or what we call “controlling for prior knowledge”, L.223 for scenario 1, L.235 for scenario 2, and  L.245 for scenario 3, and we give more explanations about it in section B.1 of the SI. We will further clarify that in the paper.

---

> > ### Comment · Reviewer_LEiC · 2022-08-06
> > **Still some concerns**
> >
> > In general most of my concerns/questions have been addressed and I have modified my score accordingly, however that are still a couple left, that I think are important
> >
> > __This comment is misguided. A reproducibility concern would arise if the methods were not described in sufficient detail to allow someone else to run the same exact set of experiments. This is not the case here since we will provide the code and all necessary details. Filtering participants for online experiments is standard and necessary to remove participants that are not properly engaged or following explanations [1,2]. We agree that the use of human participants creates a burden on the researcher over automated scores but given that our results suggest that current scores are sometimes misleading we do not see any alternatives at the moment. We would like to highlight that this framework is one of the very few human studies [3] that can be scaled, as all code will be made available and all types of explainability methods can be tested.__
> >
> > >Yes, my initial comment was misguided, I did not phrase correctly, I'll try to explain myself better now. My "reproducibility" concern is more of a problem with the fact that tuning any "new" explainer with this setup is problematic, since everytime I want to change a hyper-parameter I would have to re-run the entire human experiment so it is very hard to adapt an explainer to maximize the proposed metric in the benchmark.  However, _I do understand this is an intrinsic limitation of this kind of setup_.  This is why I proposed to rebrand the paper as a study of attribution methods.
> >
> > __We are also interested in evaluating counterfactuals on the use case where attribution methods are not useful. But this would constitute future work because our main focus is to show that attribution methods do not help because they lack the “what” component, which is independent of whether or not current counterfactual explainers address this concern.__
> >
> > >My concern here is that, the main conclusion of the paper is something we already know which is that attribution methods have severe limitations, because they lack the "what" component as mentioned in [1] _"Heatmaps are useful to understand things like which objects in an image contributed to a classification. However, they cannot visualize/explain well attributes that are not spatially localized, like size, color, etc. In addition, they can show which areas of the image may be changed in order to affect the classification, but not how they should be changed."_
> >
> > __Having to classify more than 2 classes would increase the participants’s cognitive load and bring unnecessary difficulty to the task. This is not necessary because classification problems with more than 2 classes can always be reformulated as one vs. all / binary classification problems without lack of generality. Our task could be trivially reformulated as predicting Target class vs Other, instead of Class 1 vs Class 2 as we did in the paper...__
> >
> > >Yes that is precisely my point, the fact that the meta-predictor is a human, really limits the amount of classes with which the benchmark will work.  Consider scenario 3 where all the classes being evaluated are really similar, even in a one vs all setting I seriously doubt a human would perform well in that circumstance given a high amount of classes (10, 100).
> >
> >
> > [1] Lang, O., Gandelsman, Y., Yarom, M., Wald, Y., Elidan, G., Hassidim, A., ... & Mosseri, I. (2021). Explaining in style: Training a gan to explain a classifier in stylespace. In Proceedings of the IEEE/CVF International Conference on Computer Vision (pp. 693-702).

---

> > > ### Author Response · Authors · 2022-08-09
> > > **RE: Still some concerns**
> > >
> > > We are pleased to have successfully addressed most of the reviewer's concerns. Please kindly find additional clarifications below.
> > >
> > > **Yes, my initial comment was misguided, I did not phrase correctly, I'll try to explain myself better now. My "reproducibility" concern is more of a problem with the fact that tuning any "new" explainer with this setup is problematic, since everytime I want to change a hyper-parameter I would have to re-run the entire human experiment so it is very hard to adapt an explainer to maximize the proposed metric in the benchmark. However, I do understand this is an intrinsic limitation of this kind of setup. This is why I proposed to rebrand the paper as a study of attribution methods.**
> > >
> > > >We want to make it clear that we are indeed not proposing a metric to optimize every hyper-parameter of the explainer, but rather what we see as the first robust human-centered method to evaluate how good explainability methods are at explaining how models behave. With that said, since the reviewer’s main concern seems to have to do with “branding” can we suggest a new title as a compromise? What I Cannot Predict, I Do Not Understand: A Human-Centered Approach to Evaluate Explainability Methods and an Application to SOTA Attributions Methods.
> > >
> > > **My concern here is that, the main conclusion of the paper is something we already know which is that attribution methods have severe limitations, because they lack the "what" component as mentioned in [1] "Heatmaps are useful to understand things like which objects in an image contributed to a classification. However, they cannot visualize/explain well attributes that are not spatially localized, like size, color, etc. In addition, they can show which areas of the image may be changed in order to affect the classification, but not how they should be changed."**
> > >
> > > >Two points: We want to emphasize that while we agree that the possible limitations of attribution methods might have been discussed before in the literature, 1) there are still dozens of attribution methods that are being developed every year, and 2) our study is the first to demonstrate these limitations scientifically by identifying positive and negative test scenarios (in addition to laying the groundwork for a potential image-computable metric to identify those scenarios).
> > >
> > > **Yes that is precisely my point, the fact that the meta-predictor is a human, really limits the amount of classes with which the benchmark will work. Consider scenario 3 where all the classes being evaluated are really similar, even in a one vs all setting I seriously doubt a human would perform well in that circumstance given a high amount of classes (10, 100).**
> > >
> > > >Again we believe this is a misunderstanding of our approach.
> > > If the concern of the reviewer has to do with going beyond binary datasets, the problem can be trivially reformulated as a Target class vs Other binary classification problem. As the human only has to learn the rules the model uses for the Target class, he should be able to perform well regardless of the number of classes in the dataset.
> > > If the concern of the reviewer has to do with human cognitive limitation, we believe the reviewer’s concern is unwarranted. What we propose is a human-derived metric, therefore, we do not understand why accurately reflecting human cognitive limitation is a “limit” of the proposed evaluation method. If a participant does not perform well in a setting, it should be an accurate reflection of the performance of end-users in such setting.

---

> > > > ### Comment · Reviewer_LEiC · 2022-08-09
> > > > **Updated my score**
> > > >
> > > > The major concerns I've raised have been sucessfully adressed by the authors, thus I am updating my score accordingly.

---

### Official Review · Reviewer_HHXE · 2022-07-08

**Rating:** 6
**Confidence:** 3
**Soundness:** 3 good
**Presentation:** 3 good
**Contribution:** 3 good

**Summary:**

The authors propose a quantitative measurement to evaluate explainability methods by relying on how well the methods allows a human to predict what prediction a classifier will make. The authors conduct an experiment using their approach for a variety of image classification explainability tools.

**Questions:**

1. Might this approach apply for other definitions of an effective explanation?
2. Is this actually a psychophysics experiment?

**Limitations:**

The authors should deeply engage with the limitations around how creating a benchmark for their narrowly defined idea of an effective explanation might cause harm by shifting focus away from other ones.

**Strengths And Weaknesses:**

Strengths:
Some sort of statistical approximation around the effectiveness of human-facing explanations is a worthwhile, if challenging goal.
The methods the authors propose, assuming one buys their narrow definition of an effective explanation, seems reasonable and defensible.
The initial experiments produce interesting and potentially useful results.
The paper is reasonably easy to follow, though would benefit from making fewer assumptions about reader's background knowledge in the introduction.

Weaknesses:
The concept of a meta-predictor seems to limit the author's approach to problems in which a model's goal is simply to replicate a human's intelligence, rather than to augment it or enhance it in ways the human could not do on its own.
The authors claim they conducted psychophysics experiments. In what ways does their study design engage with psychophysics? The authors should explicitly say what they are borrowing from the psychophysics literature and why. (From my understanding of psychophysics, the connections seem tenuous and superficial at best).

---

> ### Author Response · Authors · 2022-08-02
> **Response to reviewer HHXE**
>
> Thank you for your positive feedback and interesting questions! Please kindly find our replies below.
>
> **The methods the authors propose, assuming one buys their narrow definition of an effective explanation, seems reasonable and defensible.**
>
> >We argue that our definition is anything but narrow. We define explainability methods broadly as methods that are designed to help users better understand their model. Our evaluation method, which we call “usefulness” measures exactly this, how much they help. Furthermore, we emphasize L.47 that this definition of an effective explanation is backed-up by a body of literature on the psychology of explanations. We would gladly add a paragraph in the discussion if the reviewer could please expand on what would be broader definitions of effective explanations?
>
> **The concept of a meta-predictor seems to limit the author's approach to problems in which a model's goal is simply to replicate a human's intelligence, rather than to augment it or enhance it in ways the human could not do on its own**
>
> >We think the reviewer may have misunderstood our approach. By “meta-predictor”, we mean being able to understand a model sufficiently well to be able to predict how it will classify an image. There is absolutely no requirement for the underlying strategy to match that of a human observer as long as it is comprehensible to a human. In fact, our “leaves” dataset corresponds to a use case for which humans do not come equipped with any a priori strategy [1].
>
> **The authors claim they conducted psychophysics experiments. In what ways does their study design engage with psychophysics? The authors should explicitly say what they are borrowing from the psychophysics literature and why. (From my understanding of psychophysics, the connections seem tenuous and superficial at best).**
>
> >We conducted human behavioral experiments where we quantitatively measured the response of participants to the presentation of visual stimuli. These experiments required approval from an Institutional Review Board (IRB) and can be unambiguously labeled as psychophysics.
>
> **Might this approach apply for other definitions of an effective explanation?**
>
> >This is related to an earlier point made by the reviewer. We are having trouble imagining alternative definitions. Could the reviewer please be more specific about what they have in mind for an alternative definition?
>
> **The authors should deeply engage with the limitations around how creating a benchmark for their narrowly defined idea of an effective explanation might cause harm by shifting focus away from other ones.**
>
> >Again this is related to the above points. We would need more details about alternative definitions of effective explanations to answer this point.
>
> * [1] Edward J Spagnuolo, Peter Wilf, and Thomas Serre. “Decoding family-level features for modern and 506 fossil leaves from computer-vision heat maps”. In: American journal of botany (2022)

---

> > ### Comment · Reviewer_HHXE · 2022-08-09
> > **Maintain my score**
> >
> > Thank you to the authors for their response to my review. Overall I am maintaining my score of weak accept.
> >
> > * effective explanation definition: the paper's methods defines an effective explanation as one that enables people to “predict the predictor”. Other potential definitions would involve the user being able to accurately explain how the algorithm works, not just the predictions it will make.
> >
> > * psychophysics experiments: I agree this falls under the broadest definition of a psychophysics task, but typically when a researcher explicitly defines their experiment as a psychophysics experiment, it is good practice to place their experiment within the psychophysics literature (e.g. does it build off a classical psychophysical method?)

---

> > > ### Author Response · Authors · 2022-08-09
> > > **RE: Maintain my score**
> > >
> > > We thank the reviewer for reading our review. Please kindly find further clarifications below.
> > >
> > > **effective explanation definition: the paper's methods defines an effective explanation as one that enables people to “predict the predictor”. Other potential definitions would involve the user being able to accurately explain how the algorithm works, not just the predictions it will make.**
> > >
> > > >We believe that while the potential definition given by the reviewer is sound, in theory, it is not actionable in practice. To evaluate if users “accurately explain how the algorithm works” we would need a ground truth explanation about how the algorithm works, which we do not have for deep neural networks. Our definition is narrower but it is quantifiable using our framework.

---

### Official Review · Reviewer_jUUQ · 2022-07-10

**Rating:** 6
**Confidence:** 3
**Soundness:** 3 good
**Presentation:** 4 excellent
**Contribution:** 3 good

**Summary:**

This paper addresses evaluating explanation methods for image classification based on human experiments. To evaluate them, this study proposes an evaluation metric that quantifies how well humans are able to understand the output of the system after looking at the explanation provided by each explanation method. Then, this study analyzes the explanation methods by considering three scenarios to do bias detection, identification of new strategies, and understanding of failure cases.

**Questions:**

NA

**Strengths And Weaknesses:**

## Strengths
- The experiment is well designed, the hypotheses tested are appropriate, and the experiment is done with enough participants. Therefore, the results obtained look convincing, except for the points in the weaknesses below.
- Some results are valuable for XAI community. For example, the result of the third scenario can strengthen the statement in previous works, e.g., [57].

## Weaknesses
- I think the impact of the results is limited because the experiment is conducted only on limited scenarios, such as classification on Husky vs Wolf and classification on Kit Fox and Red Fox. Due to the limitation, the scores shown in Figures 4-6 are distributed in narrow areas. Therefore, if more scores are obtained by conducting experiments on more scenarios, the results may change.
- There is a possibility that the predictive performance of the recognition model affects the utility score. When the recognition model has a poor predictive performance, the explanation methods depending on the predictions by the recognition model may also produce incorrect and uncertain explanations. The three scenarios in this study are settings that are difficult for the recognition model to accurately classify images. Therefore, I have a concern that whether the experiment can evaluate explanation methods only, excluding the impact of the recognition model.

---

> ### Author Response · Authors · 2022-08-02
> **Response to reviewer jUUQ**
>
> Thank you for your positive feedback and valuable comments. Please kindly find our replies below.
>
> **I think the impact of the results is limited because the experiment is conducted only on limited scenarios, such as classification on Husky vs Wolf and classification on Kit Fox and Red Fox. Due to the limitation, the scores shown in Figures 4-6 are distributed in narrow areas. Therefore, if more scores are obtained by conducting experiments on more scenarios, the results may change.**
>
> >We would like to point out that the number of scenarios considered in our paper (=3 is greater than for any of the prior related work (respectively 1 scenario [1] and 2 scenarios [2,3,4]). The scenarios we chose are representative of different use cases: Husky vs Wolf is a classic example of bias detection [5]; Kit Fox vs Red Fox is a reasonable example of failure cases on a fine-grained dataset and the “leaves” dataset is a non-trivial use case for which humans do not come equipped with any a priori strategy and where they can potentially benefit from explanation methods to understand the underlying strategy used by a model [6]. We would also like to emphasize that all our experiments were conducted on a sufficiently large number of participants for the results to be statistically reliable. Adding more experiments or scenarios would not change the main contributions of the paper. That is, (i)  a novel human-derived metric to evaluate the usefulness of explanations, (ii) a corresponding large study of attribution methods, and (iii) evidence that the widely used faithfulness metrics are unable to predict failure cases of attribution methods but that perceptual similarity scores derived from the explanations can do a better job at it.
>
> **There is a possibility that the predictive performance of the recognition model affects the utility score. When the recognition model has a poor predictive performance, the explanation methods depending on the predictions by the recognition model may also produce incorrect and uncertain explanations. The three scenarios in this study are settings that are difficult for the recognition model to accurately classify images. Therefore, I have a concern that whether the experiment can evaluate explanation methods only, excluding the impact of the recognition model.**
>
> >This is indeed a very important concern, that we think was carefully taken into consideration in our methodology. The proposed evaluation method measures how well people understand the strategy of the model, whether the model uses a good/correct strategy or not. Because participants predict a model’s decision and not the true label, it does not matter whether the model is good at recognition or not, the only thing that impacts the results is how easy the strategy of the model is for participants to comprehend -- which is independent of any performance metric –, and how well the explanation method captures that strategy and translates it in a manner which is interpretable by users –which is what we want to evaluate–.
>
> * [1] Hua Shen and Ting-Hao Huang. “How Useful Are the Machine-Generated Interpretations to General Users? A Human Evaluation on Guessing the Incorrectly Predicted Labels”. In: Proceedings of the AAAI Conference on Human Computation and Crowdsourcing. Vol. 8. 1. 2020, pp. 168–172.
> * [2] Peter Hase and Mohit Bansal. “Evaluating Explainable AI: Which Algorithmic Explanations Help Users Predict Model Behavior?” In: Proceedings of the Annual Meeting of the Association for Computational Linguistics (ACL Short Papers). 2020.
> * [3] Giang Nguyen, Daeyoung Kim, and Anh Nguyen. “The effectiveness of feature attribution methods and 468 its correlation with automatic evaluation scores”. In: Advances in Neural Information Processing Systems (NeurIPS), 2021.
> * [4] Kim, S. S., Meister, N., Ramaswamy, V. V., Fong, R., & Russakovsky, O. “HIVE: Evaluating the Human Interpretability of Visual Explanations”. In: European Conference of Computer Vision (ECCV). 2022.
> * [5] Marco Tulio Ribeiro, Sameer Singh, and Carlos Guestrin. “"Why Should I Trust You?": Explaining the Predictions of Any Classifier”. In: Proceedings of the 22nd ACM SIGKDD international conference on knowledge discovery and data mining. 2016.
> * [6] Edward J Spagnuolo, Peter Wilf, and Thomas Serre. “Decoding family-level features for modern and 506 fossil leaves from computer-vision heat maps”. In: American journal of botany (2022)

---

### Official Review · Reviewer_bQ3u · 2022-07-11

**Rating:** 5
**Confidence:** 4
**Soundness:** 3 good
**Presentation:** 3 good
**Contribution:** 2 fair

**Summary:**

The paper introduces an evaluation framework for explanatory methods that uses human ability to predict a model's predictions as a measure of the utility of the explanations. The paper considers 3 scenarios for usefulness: bias identification, understanding how to solve too difficult tasks for non-experts, and understanding of failure cases. The paper looks at 6 attribution methods over 1 dataset per scenario.

**Questions:**

How do you think the evaluation method can help the community to design better explainability methods in the first place? Assume that one develops a novel explainer but evaluating it on your method shows that it is less useful than e.g., SmoothGrad. Would your evaluation provide any further insight as to how the novel explainability method could be improved to become more useful?

**Limitations:**

I did not see a direct mention of potential limitations or negative societal impact, which I suggest the authors incorporate in their paper.

**Strengths And Weaknesses:**

Strengths
The paper addresses a very important topic: the evaluation of explainability methods.
It is clearly written and has a good research methodology.
Their evaluation method is applicable to any explainability method. While the principal behind it is not novel (as the authors also point out), the evaluation is done in a more thorough manner than in the previous studies.
Some of the results provide new and unexpected insight, which could potentially have a good impact on shaping the explainability field.


Weaknesses
There isn't much novelty in the evaluation method proposed, just minor improvements over previous studies, although it is still worth having these improvements.

For each of the 3 scenarios, only 1 dataset was used.

Only attribution methods were tested. I think it is already quite clear for the community that attribution methods have lots of problems (and clearly they cannot provide the "what" that the authors mention) and therefore it would have been more interesting to also test other types of explanatory methods. But even among attribution methods, popular ones such as SHAP and LIME were not tested.

Some of the conclusions have already been established before and/or are to be expected, as the authors also point out. However, it is still useful to have this paper show them again in a proper manner (e.g., eliminating potential confounds).


Minor typos:
Use \citet when appropriate
L74 "score drop in score"
L81 remove . before citation
L85 citet [5]
L103 This removeS
L107 measure reflectS
L107 and L109, 112 explanationS
L178: Meta-predictorS
L191: space missing ".T"
L220 "ref. [5]" delete ref and citet
L229: if --> is
L320 citation missing
L328: capital for section
L367: vs. remove larger space
L394 one needS
L394: is -> are

---

> ### Author Response · Authors · 2022-08-02
> **Response to reviewer bQ3u**
>
> Thank you for the positive comments about the value of our results and the insightful questions! Please kindly find our replies below.
>
>
>
> **There isn't much novelty in the evaluation method proposed, just minor improvements over previous studies, although it is still worth having these improvements.**
>
> >We respectfully disagree. Prior work either has serious methodological issues and/or was applied to a very different domain (tabular data vs. images). To the best of our knowledge, our study is the first to describe a properly designed human psychophysics experiment to evaluate the usefulness of explanation methods. This is not a trivial contribution!
>
> **For each of the 3 scenarios, only 1 dataset was used.**
>
> >We find the reviewer’s comment unfair as we use a total of 3 datasets which is more than any of the prior related work (1 dataset [1] and 2 datasets [2,3,4]).
>
> **Only attribution methods were tested. I think it is already quite clear for the community that attribution methods have lots of problems (and clearly they cannot provide the "what" that the authors mention) therefore it would have been more interesting to also test other types of explanatory methods.**
>
> >We focused on attribution methods because they are currently the most popular ones. Yes it is known that they have problems but we are not aware of any prior work systematically studying their limitations. To the best of our knowledge, our study is the first to rigorously study their pitfalls. We even propose a novel metric that can help identify cases where attribution methods will most likely not be useful. Our approach is general and should in principle be applicable to other explainability methods.
>
> **But even among attribution methods, popular ones such as SHAP and LIME were not tested.**
>
> >We agree that SHAP and LIME used to be popular methods, but they are no longer considered competitive on natural images, and are no longer used in recent studies (e.g., see refs [3, 5] both NeurIPS 2021). We used Occlusion, Grad-CAM, Saliency, SmoothGrad, Integrated Gradient and Input Gradient, which are commonly used as benchmarks in recent papers [5,6,7].
>
> **How do you think the evaluation method can help the community to design better explainability methods in the first place? Assume that one develops a novel explainer but evaluating it on your method shows that it is less useful than e.g., SmoothGrad. Would your evaluation provide any further insight as to how the novel explainability method could be improved to become more useful?**
>
> >It is true that our evaluation method only allows for a comparison with other methods but in all fairness there is currently no method that can be used for this purpose. Automated measures suffer from the same limitations, and on top of it, we show in this work that greater faithfulness does not necessarily mean greater human interpretability. We have added this point in the limitations.
>
> **I did not see a direct mention of potential limitations or negative societal impact, which I suggest the authors incorporate in their paper.**
>
> >We have incorporated a section about limitations and societal impact in SI.
>
> * [1] Hua Shen and Ting-Hao Huang. “How Useful Are the Machine-Generated Interpretations to General Users? A Human Evaluation on Guessing the Incorrectly Predicted Labels”. In: Proceedings of the AAAI Conference on Human Computation and Crowdsourcing. Vol. 8. 1. 2020, pp. 168–172.
> * [2] Peter Hase and Mohit Bansal. “Evaluating Explainable AI: Which Algorithmic Explanations Help Users Predict Model Behavior?” In: Proceedings of the Annual Meeting of the Association for Computational Linguistics (ACL Short Papers). 2020.
> * [3] Giang Nguyen, Daeyoung Kim, and Anh Nguyen. “The effectiveness of feature attribution methods and 468 its correlation with automatic evaluation scores”. In: Advances in Neural Information Processing Systems (NeurIPS), 2021.
> * [4] Sunnie SY Kim et al. “HIVE: Evaluating the Human Interpretability of Visual Explanations”. In: European Conference of Computer Vision (ECCV). 2022.
> * [5] Fel Thomas et al. “Look at the Variance! Efficient Black-box Explanations with Sobol-based Sensitivity Analysis”. In: Advances in Neural Information Processing Systems (NeurIPS). 2021.
> * [6] Adebayo, J., Gilmer, J., Muelly, M., Goodfellow, I., Hardt, M., & Kim, B. (2018). Sanity checks for saliency maps. Advances in neural information processing systems, 31.
> * [7] Fel, T., Ducoffe, M., Vigouroux, D., Cadène, R., Capelle, M., Nicodème, C., & Serre, T. (2022). Don't Lie to Me! Robust and Efficient Explainability with Verified Perturbation Analysis. In ICML: Workshop on Formal Verification of Machine Learning (2022)

---

> > ### Comment · Reviewer_bQ3u · 2022-08-09
> > **Answer to rebuttal**
> >
> > Thank you for the answers. I am slightly improving my score, however, the argument that a couple of other papers (one of which is a short paper -- hence acceptable to have fewer datasets) are using 2 datasets is not a convincing one. Moreover, while you have 3 datasets in total, each scenario has only one, which is also very trivial in some cases, e.g., Huskey vs Wolf. Would this method help with more complex datasets? (this is also the concern of Reviewer LEiC). More (and more complex) datasets per scenario are needed to prove that the work would be helpful in practice.

---

> > > ### Author Response · Authors · 2022-08-09
> > > **RE: Answer to rebuttal**
> > >
> > > We thank the reviewer for taking into consideration our comments and raising his score.

---

### Meta-Review · Area_Chair_9Rew · 2022-08-23

**Recommendation:** Accept
**Confidence:** Certain

**Metareview:**

This paper introduces a human evaluation framework for benchmarking current explainers.

There was an engaged discussion between authors and reviewers. Many concerns were clarified and the average score was raised from 4.75 to 5.5. Some concerns remain regarding the intrinsic limit of human evaluation, but overall, there are no major flaws with the current study and the framework is likely to be valuable since human evaluation of explainers is central to the progress of XAI.

Hence, I recommend acceptance of the paper.


**Award:**

No

---

### Decision · Program_Chairs · 2022-09-14

Accept